

# EnKF with closed-eye period - towards a consistent aggregation of information in soil hydrology

Hannes H. Bauser[1,2], Stefan Jaumann[1,2], Daniel Berg[1,2], and Kurt Roth[1,3]

[1]Institute of Environmental Physics (IUP), Heidelberg University, Heidelberg
[2]HGS MathComp, Heidelberg University, Heidelberg
[3]Interdisciplinary Center for Scientific Computing (IWR), Heidelberg University, Heidelberg

*Correspondence to:* Hannes H. Bauser (hannes.bauser@iup.uni-heidelberg.de)

**Abstract.** The representation of soil water movement exposes uncertainties in all model components. We assess the key uncertainties for the specific hydraulic situation of a 1D soil profile with TDR measured water contents. Addressed uncertainties are initial condition, soil hydraulic parameters, small scale heterogeneity, upper boundary condition, and the local equilibrium assumption by the Richards equation. We employ an ensemble Kalman filter (EnKF) with an augmented state to represent and estimate all key uncertainties, except for the intermittent violation of the local equilibrium assumption. For the latter, we introduce a Closed-Eye EnKF to bridge the gap. Due to an iterative approach, the EnKF was capable to estimate soil parameters, Miller scaling factors, and upper boundary condition based on TDR measurements during a single rain event. The introduced closed-eye period ensured constant parameters suggesting that they resemble the believed true material properties. This closed-eye period improves predictions during periods when the local equilibrium assumption is met, but requires a description of the dynamics during local non-equilibrium phases to be able to predict them. Such a description remains an open challenge. Finally, for the given representation our results show the necessity to include small scale heterogeneity. A simplified representation with Miller scaling already yielded a satisfactory description.

## 1 Introduction

The description of soil water flow in the vadose zone with a mathematical model requires knowledge about material properties (typically characterized by soil hydraulic parameters), initial conditions, and boundary conditions. Especially the material properties are difficult to determine, since they can neither be measured directly nor transferred directly from the laboratory to the field.

Soil hydraulic parameters have been estimated inversely based on measurements of the temporal development of the hydraulic state with reviews e.g. by Hopmans et al. (2002) and Vrugt et al. (2008). Inversions are based on a perfect model assumption and incorporate all errors and uncertainties into the estimated parameters, leading to a suboptimal estimation.

In contrast, data assimilation methods are capable to combine information from measurements and models into an optimal estimate of the geophysical field of interest, but depend on the correct description of corresponding uncertainties (Reichle, 2008). Data assimilation methods include variational methods, particle filters, and ensemble Kalman filters (EnKF).

The EnKF, introduced by Evensen (1994), became popular in hydrology and was used to estimate e.g. water content states





from satellite data (e.g. Reichle et al., 2002) or local water content measurements (e.g. De Lannoy et al., 2009).

Liu and Gupta (2007) reminded that in hydologic data assimilation the focus on state estimation is not sufficient and called for the incorporation of the errors in model structures and parameters.

Moradkhani et al. (2005) used a dual EnKF approach, whereas Vrugt et al. (2005) combined an EnKF and the shuffled complex
evolution metropolis algorithm in an iterative way, to estimate parameters and states in a rainfall-runoff model. Chen and Zhang (2006) used an EnKF with augmented state to simultaneously estimate pressure and conductivity field for saturated flow.

Based on the Richards equation, Li and Ren (2011) first employed the simultaneous estimation of states and soil hydraulic parameters with the EnKF, which has been studied in more detail for purely synthetic cases (e.g. Wu and Margulis, 2011; Song et al., 2014; Erdal et al., 2015; Shi et al., 2015; Man et al., 2016) or with focus on synthetic cases (e.g. Wu and Margulis, 2013;
Erdal et al., 2014). The advantage of synthetic cases comprises the direct control and knowledge about all uncertainties.

In this study, we focus on a real world case to address the challenge of consistent aggregation of the information, which requires a proper characterization of uncertainties in all model components. We exercise this aggregation with the EnKF on a small situation: a 1D soil profile equipped with time domain reflectometry (TDR) probes measuring water content, during a time period of less than 2 months.
Our goals are (i) to assess all uncertainties in the representation of this particular situation qualitatively and (ii) to reduce the largest uncertainties with the EnKF or to consider them appropriately.

## 2 Background

This section describes our understanding of (i) a representation of a physical system in general, (ii) the hydrological system we want to represent, (iii) the specific representation of this system in detail, (iv) the concept of knowledge fusion and (iv) the
EnKF.

### 2.1 Respresentation

In this work, we call the mathematical description of a physical system a representation, which comprises in the most general sense the four components: dynamics, forcing, subscale physics and states.

**Dynamics:** The dynamics specifies how the state of the system is propagated in space and time at the scale of interest.
**Forcing:** The forcing or coupling of the system to the superscale physics in space (initial state) and time (boundary condition).
**Subscale pyhsics:** The subscale physics not explicitly described by the dynamics, but with a parameterization comprising one or more parameters and their spatial and temporal distribution.
**State:** Distribution of the variable propagated by the dynamics, at a specific time. Notice that this excludes parameters.

According to this definition a representation is specific for each system. Hence, we first introduce the system, and the details
of the representation afterwards.





## 2.2 Testsite

We aim to represent the water movement in a soil profile at the Grenzhof test site close to Heidelberg, Germany. Since 2003, experiments are conducted at the test site. In 2004, a weather station was built, which measures precipitation and further atmospheric data (wind, temperature, incoming and outgoing long- and shortwave radiation, relative humidity and air temperature).

A detailed description of the test site can be found in Wollschläger et al. (2009).

In 2009, a soil profile was equipped with 11 TDR probes measuring water content hourly. The soil profile itself, described explicitly by Schenk (2011), is depicted in Fig. 1. It consists of four different soil layers up to the depth of 1.4 m. The top three layers are equipped with three TDR probes each, and two TDR probes in the bottom layer. The profile is kept free from vegetation at the surface.

The complete time period considered comprises 60 days from October 1, 2011 (day 1) until November 29, 2011 (day 60). The boundary condition along with the water content measured by the topmost TDR are shown in Fig. 2.

## 2.3 Respresentation of soil water flow

For this specific soil profile we can now formulate the representation consisting of the four components described in section 2.1. For each we give the a priori description, as well as the corresponding assumptions and sources of uncertainty.

**Dynamics:** For a continuous porous medium, the Richards equation (Richards, 1931) describes the change of volumetric water content $\theta$

$$\frac{\partial \theta}{\partial t} - \nabla \cdot \left[ K(\theta) \left[ \nabla h_m(\theta) - 1 \right] \right] = 0, \tag{1}$$

with the isotropic conductivity $K$ and matric head $h_m$. The Richards equation itself is process-based and is expected to de-

scribe the actual physics adequately as long as its underlying assumptions are fulfilled: It assumes the soil water movement to be a single-phase process. Hence, the Richards equation is expected to fail in very dry or wet conditions. Both are not observed at the Grenzhof test site during the time period considered. The corresponding uncertainties are small. The Richards equation further assumes local equilibrium. This is assumed to be violated during strong infiltration events, e.g., during heavy rainfall. We expect local non-equilibrium at the Grenzhof test site, but it is a priori not clear, how strong the forcing has to be or how to

describe these cases. This creates uncertainty in the dynamics during rain events.

We additionally assume that horizontal flow is negligible and that we can describe the system one dimensional without additional sources or sinks. The flat terrain and horizontal layering at the Grenzhof test site supports this assumption. Still, heterogeneity, that is not represented, might introduce two-dimensional flow.

**Material properties:** The subscale physics not explicitly represented in Eq. 1, are the dependencies of $K$ and $h_m$ on $\theta$. Several parameterizations are available. We follow Wollschläger et al. (2009) and choose the Mualem-van Genuchten parame-





terization (Mualem, 1976; Van Genuchten, 1980):

$$K(\Theta) = K_0 \Theta^\tau \left[ 1 - \left[ 1 - \Theta^{n/[n-1]} \right]^{1-1/n} \right]^2, \tag{2}$$

$$h_m(\Theta) = \frac{1}{\alpha} \left[ \Theta^{-n/[n-1]} - 1 \right]^{1/n}, \tag{3}$$

with the saturation

$$\Theta := \frac{\theta - \theta_r}{\theta_s - \theta_r}. \tag{4}$$

The parameterization reduces the description of the subscale physics to a set of six parameters: $\theta_s$, $\theta_r$, $\alpha$, $n$, $K_0$ and $\tau$. We do not expect significant uncertainties due to assuming this particular parameterization, because we only observe a rather small water content range.

The parameters $\theta_s$ and $\theta_r$ are the saturated and residual water content. The parameters $\alpha$ and $n$ can be associated with an inverse air entry value and the width of the pore size distribution respectively, although there is no direct physical relation. $K_0$ is the saturated conductivity and $\tau$ is a fudge factor.

The spatial distribution of these material properties has to be defined on the scale of the dynamics. The large-scale structure for this soil profile are 4 soil layers, where we assign an individual parameter set to each layer.

Although these parameters cannot be measured directly and are often assigned with the largest uncertainties in soil hydrology, we can in this case use the soil parameters determined by Wollschläger et al. (2009) through an inversion for a soil profile close by. They are expected to differ from the soil hydraulic parameters at the represented soil profile due to spatial heterogeneity.

We do not include a description of hysteresis, leading to additional uncertainties connected with the parameterization.

We assume the parameters to be constant in time, which appears justified, since we only investigate a short time period of less than 2 months.

The assumption of homogeneous soil layers is presumably wrong. A possibility to describe the texture inside soil layers in a simplified way is Miller scaling (Miller and Miller, 1956). They assume geometrical similarity and scale the material properties at each location according to:

$$K(\theta) = K^*(\theta) \cdot \xi^2, \tag{5}$$

$$h_m(\theta) = h_m^*(\theta) \cdot \frac{1}{\xi}, \tag{6}$$

where $^*$ denotes the reference material properties for each layer and $\xi$ is the Miller scaling parameter. The assumption of geometric similarity does not necessarily hold and leads to some uncertainties. The scaling factor themselves are a priori unknown. They are assumed to be 1, which introduces large uncertainties.

**Forcing:** The forcing in time is the upper and lower boundary condition. For the upper boundary condition, we can measure rain and use the meteorological data from the on-site weather station to calculate the evaporative water flux using the reference FAO Penman-Monteith equation following Allen et al. (2006). Here, especially the estimation of the evaporation is highly




uncertain. The daily mean value can differ 10-20% (Foken, 2006). In dry conditions the evaporation is limited by the soil, which leads to additional uncertainties.

At the Grenzhof test site the rain gauge had not been calibrated during the period of consideration. Because of that we have to assume an uncertainty of about 20% for the precipitation as well.

For the lower boundary condition there are no measurements at the site. However, constant water content measurements in the lowest soil layer during the observed time period indicate, that the dynamics is decoupled from the groundwater table. Therefore, we follow Wollschläger et al. (2009) and add a 5th layer with soil parameters for sand from Carsel and Parrish (1988) up to a depth of 4 m. The sand layer decouples the dynamics from the groundwater table, that we keep constant at the depth of 4 m. This leads to uncertainties, but they are expected to be negligible.

The forcing or embedding in space is the initial condition. The initial condition is difficult to estimate. A simple approach is to use the measured water contents and interpolate them linearly with constant extrapolation to layer boundaries. This leads to large uncertainties in between.

**States:** With the three components dynamics, material properties and forcing we can predict the temporal development of
the states, in this case the water content. These states will have large uncertainties due to the uncertainties in the other components.

An additional source of uncertainty is the simulation of the dynamics with a numerical solver, here with MuPhi (Ippisch et al., 2006). The assumption of local equilibrium in the Richards equation has to be granted on the grid size of the numerical
model as well. Since we have a very small test case, we choose a resolution of 1 cm to minimize the effect.

Water content determined with TDR probes will be used to improve the representation. The water content values are calculated from the temperature corrected electric permittivity with the complex refraction index model (CRIM) following Roth et al. (1990). The porosities required there, were obtained from volumetric soil samples.
We assume an unbiased uncertainty of 0.01 for the water contents. This is the largest noise observed with the TDR probes. The assumptions made for the representation of the soil water movement at the Grenzhof test site are summarized in Table 1. Along we give our qualitative assessment of the uncertainties: small, no need to represent the uncertainty explicitly; intermediate, it might be necessary to represent these uncertainties, we decide not to but keep them in mind; large, uncertainties must be represented with the goal to reduce them. Our findings depend on this uncertainty assessment and might be affected if is does
not hold.

### 2.4 Knowledge Fusion

We define knowledge fusion as the consistent aggregation of all information pertinent to some observed reality. In the presented situation at the Grenzhof test site this would require the quantitatively correct description of all uncertainties in the representation and a subsequent optimal reduction of all these uncertainties based on all additionally available information. For the





measurement part these are primarily the water content data from the 11 TDR probes, but any other information, even expert knowledge, should be incorporated as well. So far, this goal is only partly feasible, though.

With an inversion all information and uncertainties would be included into the parameters, which is not a consistent aggregation of the information, since the structural errors in the other components are not represented.

5 Data assimilation methods are capable to represent all uncertainties and have been expanded to not only estimate states, but also parameters. For an aggregation of all information we have to reduce uncertainties in all representation components. In this study we aim to describe and reduce the uncertainties classified as large in Table 1 and decided to neglect the ones classified as small or intermediate with the EnKF.

## 2.5 Ensemble Kalman Filter

10 The EnKF is a data assimilation method that uses a Monte Carlo approach for an optimal state estimation, based on the assumption of unbiased Gaussian error distributions. The EnKF incorporates the measurements sequentially by alternating between a forecast step (superscript $f$), which propagates the state in time, and an analysis step (superscript $a$), which incorporates the information from the measurements at this time to improve the state.

These two steps are now explained in more detail - specifically for the given representation. A general description can be found 15 e.g in Burgers et al. (1998).

**Forecast:** First an ensemble of water content states $\boldsymbol{\theta}^i$ is propagated forward in time from k-1 to k:

$$\boldsymbol{\theta}_k^{f,i} = M(\boldsymbol{\theta}_{k-1}^{a,i}, \boldsymbol{\phi}_k^i, \boldsymbol{\xi}_k^i, \mu_k^i) + \boldsymbol{\beta}_k^i \tag{7}$$

where $M$ denotes the propagation with the Richards equation and $\boldsymbol{\phi}^i$, $\boldsymbol{\xi}^i$ and $\mu^i$ are soil hydraulic parameters, Miller scaling parameters, and the upper boundary condition of ensemble member $i$, respectively. In this way, all the uncertainties from $\boldsymbol{\theta}$, 20 $\boldsymbol{\phi}$, $\boldsymbol{\xi}$ and $\mu$ are explicitly represented and propagated forward in time with the non-linear dynamics leading to the forecast ensemble $\boldsymbol{\theta}^f$, which represents these uncertainties. Uncertainties not explicitly represented are incorporated into the process noise $\boldsymbol{\beta}$, which is difficult to describe, however. We set the process noise to 0. In this way, we represent all the uncertainties classified as 'large' in Table 1 directly, except for uncertainties caused by the local equilibrium assumption, which is expected to only hold outside of strong rain events. We decide to not represent 'intermediate' and 'small' uncertainties.

25 The uncertainty of the forecast ensemble can be characterized based on the Gaussian assumption with the state error covariance matrix $\mathbf{P}$:

$$\mathbf{P}_k^f = \frac{1}{N-1} \sum_{i=1}^N \left(\boldsymbol{\theta}_k^{f,i} - \overline{\boldsymbol{\theta}_k^f}\right) \left(\boldsymbol{\theta}_k^{f,i} - \overline{\boldsymbol{\theta}_k^f}\right)^T, \tag{8}$$

with the number of ensemble members $N$ and the ensemble mean $\overline{\boldsymbol{\theta}}$.

**Analysis:** To incorporate measurements, we have to quantify their uncertainties by defining the measurement error covariance 30 matrix $\mathbf{R}$:

$$\mathbf{R}_k = \frac{1}{N-1} \sum_{i=1}^N \left(\boldsymbol{\epsilon}_k^i\right) \left(\boldsymbol{\epsilon}_k^i\right)^T, \tag{9}$$





with the measurement errors $\epsilon$. We assume the measurement errors to be uncorrelated and drawn from a normal distribution with the assumed TDR standard deviation of 0.01.

The Kalman gain $\mathbf{K}$ weights the forecast state error covariance and the measurement error covariance

$$\mathbf{K}_k = \mathbf{P}_k^f \mathbf{H}_k^T \left[ \mathbf{H}_k \mathbf{P}_k^f \mathbf{H}_k^T + \mathbf{R}_k \right]^{-1}, \tag{10}$$

with the measurement operator $\mathbf{H}$ mapping from the state space to the measurement space, and the Kalman gain $\mathbf{K}$ mapping from the measurement space back to the state space - weighted with the uncertainties of measurements and states. Based on the Kalman gain the weighted difference of forecast and measurements is added to the forecast states to obtain the analysis states

$$\boldsymbol{\theta}_k^{a,i} = \boldsymbol{\theta}_k^{f,i} + \mathbf{K}_k \left[ \mathbf{z}_k - \mathbf{H}_k \boldsymbol{\theta}_k^{f,i} + \boldsymbol{\epsilon}_k^i \right], \tag{11}$$

where $\mathbf{z}$ are the measurements. The measurement errors $\epsilon$ have to be added to the difference so that the ensemble describes
the analysis error covariance correctly (Burgers et al., 1998). The descriptive explanation is that the measurements $\mathbf{z}$ already include measurement errors, while the measurements of the forecast state $\mathbf{H}\boldsymbol{\theta}^f$ do not. Hence, the error has to be added to compare equivalents.

By alternating between forecast and analysis, the information of all measurements is incorporated to achieve an improved estimation of the states at each time step.

By replacing the water content state $\boldsymbol{\theta}$ with an augmented state $\boldsymbol{\psi}$, not only the water content states, but all components of the augmented state can be estimated with the EnKF. We choose to include all the directly represented uncertain components in the augmented state $\boldsymbol{\psi} = [\boldsymbol{\theta}, \boldsymbol{\phi}, \boldsymbol{\xi}, \mu]$ consisting of water content, soil hydraulic parameters, Miller scaling parameters and the upper boundary condition.

Adding components to the augmented state increases its dimension, which in turn requires a larger number of ensemble members, leading to a higher computational effort. To minimize this effect, we keep the added components as small as possible.

As parameters to be incorporated, we choose $\alpha$, $n$ and $K_0$ for each of the four layers as well as the saturated water content $\theta_s$ for layers 3 and 4. For the upper two layers, the measured water contents were far away from $\theta_s$ and $\theta_r$ estimated by Wollschläger et al. (2009). Thus, they are irrelevant for the estimation, as changes in $\theta_s$ and $\theta_r$ can be compensated by the other parameters.
For the bottom two layers measurements exceeded $\theta_s$. Thus, it has to be added in the augmented state. We do not include the fudge factor $\tau$, since the observed water content ranges are rather small. For other situations, incorporation of $\tau$ might improve the results, however.

Since the EnKF assumes Gaussian distributions and linear correlations, we do not directly use $K_0$ but the logarithm (to which the water content response behaves more linearly).
The dimensions of the Miller scaling factors are reduced as well. Only the scaling factors at measurement locations are added. The whole Miller scaling field is determined by linear interpolation between measurement locations inside a layer and a constant extrapolation towards layer boundaries. As the measurements yield only little information about the small-scale architecture away from their location, we expect that this assumption has only little influence on the results. Again, the logarithm





of the parameter is used in the augmented state.

The upper boundary condition is already a scalar and is added as the flux at the surface to the augmented state.

In this way, soil parameters, Miller scaling factors and the upper boundary condition are updated besides the states in the analysis.

The expansion to an augmented state changes the propagation in time. Each component needs an individual forward propagation. We assume the soil hydraulic parameters and Miller scaling factors to be constant in time. This is not possible for the upper boundary condition where the forward equation is unknown from a soils perspective. However, measurements are available to estimate the evaporation and precipitation. Hence, we assume the forward model constant until a new estimation is available. To be able to improve the boundary condition with several measurements, we reduce the temporal resolution of

precipitation measurements to only change daily and at transitions between precipitation and evaporation. This is possible due to the dissipative nature of the Richards equation. The expanded forward propagation of the augmented state is:

$$\boldsymbol{\theta}_k^f = M_\theta(\boldsymbol{\theta}_{k-1}^a, \boldsymbol{\phi}_{k-1}^a, \boldsymbol{\xi}_{k-1}^a, \mu_{k-1}^a), \tag{12}$$

$$\boldsymbol{\phi}_k^f = \boldsymbol{\phi}_{k-1}^a, \tag{13}$$

$$\boldsymbol{\xi}_k^f = \boldsymbol{\xi}_{k-1}^a, \tag{14}$$

$$\mu_k^f = \begin{cases} \mu_{\text{meas}} & \text{if a measurement is available,} \\ \mu_{k-1}^a & \text{else.} \end{cases} \tag{15}$$

The analysis information for the additional augmented state components lies in the covariance between states at measurement locations and the components. The covariance linearizes this relation that can be highly non-linear.

To reduce the impact of the linearization of non-linear dependencies, we employ damping factors (Hendricks Franssen and Kinzelbach, 2008), different for each augmented state component (Wu and Margulis, 2011) $\mathbf{d} = [\mathbf{d}_\theta, \mathbf{d}_\phi, \mathbf{d}_\xi, \mathbf{d}_u]$. The correc-

tion vector in Eq. 11 is multiplied by the corresponding factors. We choose the values $\mathbf{d} = [1.0, 0.1, 0.1, 0.1]$. The smaller the value, the better can a non-linearity be handled, but the slower will the estimated value approach the optimal value.

Due to a limited ensemble size, the EnKF will show spurious covariances between uncorrelated state components. These can be reduced by increasing the ensemble size, or - more computationally efficient - by introducing a localization (Houtekamer and

Mitchell, 2001), where external information about uncorrelated states is introduced. This can be achieved by multiplying the error covariance matrix by the correlation matrix $\rho$ (with entries from 0 to 1) with the Schur product. This alters the calculation of the Kalman gain to

$$\mathbf{K}_t = \left(\rho \circ \mathbf{P}_t^f\right) \mathbf{H}_t^T \left[\mathbf{H}_t \left(\rho \circ \mathbf{P}_t^f\right) \mathbf{H}_t^T + \mathbf{R}_t\right]^{-1}, \tag{16}$$

where the Schur product is indicated by $\circ$.

For the water content state, the covariances are reduced with increasing physical distance. We use the 5th order piecewise rational function defined by Gaspari and Cohn (1999), which is similar to a Gaussian but with local support, which introduces



a cut off length.

For the soil parameters, we can localize even stronger by only allowing covariances (entries of 1) between parameters and measurement locations in the respective soil layer and the first measurement locations in the neighboring layers. For the Miller scaling factors, only covariances to the corresponding measurement locations are used. All other entries are set to 0. Spurious

correlations and non-Gaussian distributions can lead to filter inbreeding and ultimately filter divergence (e.g. Houtekamer and Mitchell, 1998; Hamill et al., 2001). To prevent this, the ensemble can be inflated by a multiplicative factor $\boldsymbol{\lambda}$ (Anderson and Anderson, 1999):

$$\boldsymbol{\psi}^i = \boldsymbol{\lambda}\left(\boldsymbol{\psi}^i - \overline{\boldsymbol{\psi}^i}\right) + \overline{\boldsymbol{\psi}^i}. \tag{17}$$

We choose the inflation as proposed by Anderson (2009), where the inflation factor $\boldsymbol{\lambda}$ is adapted temporally and spatially

for the states. We expand this for the other augmented state components by simply summing over the inflation factors at the measurement locations weighted with the correlations.

Additionally, we iterate the whole EnKF scheme. As we assume constant forward models for the soil parameters and Miller scaling factors, we can use the final distributions after one iteration as the initial for the next. This enables us to estimate parameters even with a small damping factor and a rather short time period of data. This is especially important for the upper

boundary condition, where the time periods of constant values are short.

    The operational assumptions of the method lead to a sub-optimal estimation of the state in each time step. Furthermore, due to the non-linear dynamics, the assumption of Gaussian distributions does not hold. It is not clear how this effects the EnKF performance in detail. The Gaussian assumption leads to a linearized state update in the analysis step. This induces erroneous

updates of those state components with dominant non-linear relation between states and measurements. These errors are alleviated by employing the damping factor, which reduces the update but as a consequence also reduces the incorporation of measurement information. Furthermore, we use the largest observed measurement noise to characterize the TDR uncertainties. This leads to possibly too large measurement uncertainties, which have a similar effect as an additional damping factor.

We do not expect a strong influence from this sub-optimal state estimation on the mean value of the results for the soil hydraulic

parameters, Miller scaling factors, and upper boundary condition. However, the final value will be approached slower than in an ideal estimation. This effect can by easily circumvent by the iterative approach.

On the downside, incorporating the same measurement information several times will lead, together with the other limitations, to incorrect quantitative uncertainties. However, the spatially and temporally adaptive covariance inflation by Anderson (2009) ensures that measurements and states agree with each other within their uncertainties. This is only a partial solution, however,

as the information is only propagated through the correlations to the measurement locations. Therefore, we will not interpret the uncertainties of the ensemble in a quantitative way. Nevertheless, we do expect them to hold qualitative information that can be interpreted.





## 3   Results

The complete time period considered comprises 60 days and ranges from October 1 (day 1) until November 29 (day 60). The boundary condition along with the water content measured by the topmost TDR are shown in Fig. 2. The whole time period is separated into 4 sections: A (day 1-2), B (day 3- 17), C (18-22) and D (23-60).

Since soil parameters can only be estimated within the observed water content range and will not be valid outside of this range, a rather large rain event is desirable. On the other hand we do not represent uncertainties associated with the assumption of local equilibrium by the Richards equation, which is violated during strong rain events. Time period C (October 18-22, 2011) combines both: a rather large total rain amount (18.2 mm) and a small maximal intensity (0.7 mm in 10 minutes).

We designed a three stage approach to improve the representation by incorporating the information from the water content
measurements of the 11 TDR probes: (1) improving the prior information during time periods A and B, (2) a standard EnKF approach, and (3) a closed eye EnKF approach, both during time period C with the considered rain event. The three stages are discussed in the following, with results analyzed through forecasts in time periods B, C and D.

### 3.1   Stage 1: Improving the prior

In a uniform layer, the water content is typically higher in deeper locations. Heterogeneity can reverse this and can lead to
higher water contents above lower water contents inside a layer. Until the infiltration front of the rain event reaches these positions, the EnKF cannot distinguish between heterogeneity and an infiltration front propagating at very low speed. This can easily lead to too small saturated conductivities especially in lower soil layers that are not reached by the infiltration front. Even upper layers can require a tuned relation of the prior uncertainties of Miller scaling factors and saturated conductivity.

To avoid too small saturated conductivities, we improve the prior of the heterogeneity using the measurements from time
interval A. Prior to A, there had been one month without rainfall leading to little dynamics during A. We start with the soil parameters estimated by Wollschläger et al. (2009) and linearly interpolated initial condition (with constant extrapolation to layer boundaries). Parameters and boundary condition are kept constant, but states and Miller scaling factors are updated with an EnKF of 40 ensemble members.

As time interval A is only a 2 days period, the filter won't be able to reach constant Miller scaling values during this short time.
Therefore, we iterate 50 times over this period, which then does lead to constant scaling factors. The resulting scaling field of the improved prior is depicted in Fig. 3. It shows that there is heterogeneity within each layer that requires a representation. Especially layer 4 exhibits a strong heterogeneity, which is already observable in Fig. 1.

Additionally, a good initial state can improve the estimation. Therefore, we guide the state with the EnKF (only state es-
timation) through time period B and achieve a better representation for the initial state for time period C, than interpolating between the measurements there.




## 3.2 Stage 2: Standard EnKF

Here, we improve the representation with the following uncertain components: improved initial condition, soil hydraulic parameters estimated by Wollschläger et al. (2009), improved Miller scaling factors, and upper boundary condition from precipitation measurements and estimated with the reference FAO Penman-Monteith equation. We iterate 10 times over time period C and

improve soil parameters, Miller scaling factors and the upper boundary condition along with the states. This estimation is performed with an ensemble size of 100 members and will be referred to as the Standard EnKF.

The initial ensembles of initial condition and Miller scaling are determined as described in stage 1. For the Miller scaling factors the uncertainty is increased again compared to the estimations in the prior, since they were estimated under the assumption of fixed soil parameters. Now they have to be able to adapt to changing parameters. The uncertainty of the natural logarithm

of the scaling factors is chosen to be 0.1. For the boundary condition shown in Fig. 2, an uncertainty of 20% is assumed. The initial soil hydraulic parameters and their uncertainties are summarized in Table 2.

In order to check the improved results, we do not show the states from the last iteration but actually run another iteration, however, this time, without incorporating the measurement information. This corresponds to an ensemble of forward runs during time period C. It is a much more strict test for the quality and objectivity of the assimilation, because the states now

cannot be adjusted. Actually, allowing incorporation of measurements leads to a much better agreement. The results for the top and bottom layer are shown in Fig. 4. The graphs show that the measurements can be represented well by the ensemble, but deviations occur, mainly during the infiltration.

Already from the TDR water contents in the first layer, it becomes clear that there is heterogeneity inside the layer. The middle TDR shows the highest water content. This effect is even stronger in layer 4. Here, the water content of the TDR about 25 cm

above the next has a water content almost 0.1 larger.

The results for the estimated Miller scaling factors are shown in Fig. 3. The Miller scaling factors were adjusted along with the hydraulic parameters during this Standard EnKF approach. This is necessary, because changing parameters require adjusted Miller scaling factors as well. The largest changes occur in layer 1, where information is introduced by the water content changes during the infiltration.

The estimated soil parameters including their uncertainties are summarized in Table 3. Although the uncertainties are quantitatively incorrect, they still indicate how much information about each parameter is in the measurements. This can be seen especially well for $K_0$. The estimation of this parameters requires dynamics. Therefore, the smallest uncertainty is observed in layer 1 and increases with the layers. In layer 4 the uncertainty of $\log_{10}(K_0)$ was reduced least, but still from 1.0 to 0.4. This shows the difficulties of the EnKF. As we did not observe a change in water content, it is unlikely that any flux was

introduced during the rain event. Nevertheless, the uncertainty was reduced, which can lead to difficulties when there actually is information.

The development of the parameters $K_0$, $\alpha$ and $n$ of the first layer during the last iteration are shown in the Fig. 5-7. All three show the same feature: The value increases during the rain event and decreases before and after, but does not change significantly between initial and final values. We interpret this as follows: The assumption of local equilibrium during the





rain event is wrong, which leads to preferential flow. The infiltration is thus too fast for the actual parameters (Fig. 4). The preferential flow leads to an apparent increase in the saturated hydraulic conductivity during the rain event, which is reduced again afterwards. This also shows that our initial assumption of zero process noise is wrong. The uncertainty introduced by the dynamics is not represented.

## 3.3 Stage 3: Closed-Eye EnKF

We enhance the EnKF with a closed-eye period during times when the local equilibrium assumption of the Richards equation does not hold. During this time, soil hydraulic parameters and Miller scaling factors are not adjusted but kept constant. In contrast, the water content state is continuously updated. In this way, the state is guided on the basis of measurements through times with uncertain dynamics without incorporating the dynamics uncertainties into the parameter estimation. The estimation of soil hydraulic parameters is only performed before and after this closed-eye period.

We use an identified non-constant parameter to define the closed-eye period. We choose $K_0$, because it shows the changes most prominently and we can directly relate it to observed physics. The closed-eye period is defined as the time period between the minimum and the maximum value of $K_0$ during the last iteration with the Standard EnKF (Fig. 8).

To perform the Closed-Eye EnKF estimation, we add another 10 iterations to the previous Standard EnKF iterations, but with the closed-eye period for the first layer. We continue with the previously estimated mean values, but increase the uncertainty for the first layer again (to half the initial uncertainty) to allow that the new values can possibly deviate from the Standard EnKF results.

The additional iterations result in new soil parameters and Miller scaling factors. The scaling factors changed only marginally compared to their regular estimates (Fig. 3). Changes in the parameters of the first layer are larger (Fig. 5-7). We recognize two things: First, the values differ from each other. For $n$ and $\alpha$ the changes are rather small, but for $K_0$ the difference is significant. The closed eye estimation yields approximately half the hydraulic conductivity as before. Second, the parameters now are constant over time and do not show fluctuations any more.

This indicates that we could extract times when the Richards equation is actually valid and we are able to determine soil hydraulic parameters that resemble the believed true material properties. There is also an apparent downside, though. We cannot use the measurements during the closed-eye period to estimate the parameters. This leads to a smaller observed water content range, limiting the parameter estimation possibilities. We call this 'apparent', because the corresponding interval does not contain valid information about the modeled system in the first place.

The forecast during time period C (analogous to the Standard EnKF) is shown in Fig. 9. The performance is acceptable, but worse than the Standard EnKF (Fig. 4). This is exactly what is expected. In the Standard EnKF we incorporated errors in the dynamics into the parameters e.g. by estimating a too large hydraulic conductivity, which can represent preferential flow up to a point. By introducing the closed-eye period, we do not incorporate these errors into the parameters any more. The parameters are thus believed to be closer to the believed true material properties. Obviously, however, these parameters fail to describe rain events that lead to hydraulic non-equilibrium.

Guiding the states through the closed-eye period is a challenge. A representation of the dynamics' uncertainty would be





required to estimate optimal states. We did not do this. The adaptive covariance inflation reduces the issue by increasing the ensemble spread when measurements cannot be explained by combined measurement and ensemble uncertainty. Still, it cannot solve the problem completely, since it was not designed to follow rapid changes of the inflation factor and it is merely based on the correlations in the ensemble. These correlations are not necessarily valid any more, when a major, spatially structured

uncertainty is not represented. In our case this did not lead to complications, because the deviations are still small and could be adjusted towards the end of the closed-eye period. In cases with stronger rain (and stronger non-equilibrium) and larger deviations of representation and measurements, we expect that a description of the dynamics uncertainties becomes mandatory.

### 3.4 Prediction capabilities

We investigate the predictive capabilities during the combined time period of B and C as well as time period D.

During time period B the total amount of rain was 23.2 mm with a maximal flux of 2.5 mm in 10 minutes (compared to a total of 18.2 mm with a maximal flux of 0.7 mm in 10 minutes for time period C). As we already assume local non-equilibrium during the rain event in time period C, we expect an even stronger effect during time period B.

The forecast results for time period B and C for soil parameters and Miller scaling parameters from the Standard EnKF and Closed-Eye EnKF are depicted in Fig. 10a and 10b (there is no improved estimation of the upper boundary condition during

time period B and we did not use the improved boundary condition from time period C). Neither of them is able to predict the water contents during the rain event in time period B in detail. However, the Standard EnKF describes the rain event in time period C better. The Closed-Eye EnKF is not able to predict the water contents during rain events, as for both local non-equilibrium is expected, which is not incorporated into the parameter estimation. For the Standard EnKF only the rain event from time period C is incorporated into the parameters and it predicts this rain event better. However, it fails to predict

the stronger rain event in time period B showing that the local non-equilibrium in different rain events affects the dynamics differently. Hence, their effects should not be included into parameters, but rather considered separately. The uncertainties are large for both EnKFs. The reason is the large uncertainty in the boundary conditions.

We emphasize that the Standard EnKF excels if the goal is to guide some partly flawed representation with a stream of data. However, if in addition it is the goal to estimate objective parameters for the representation within its range of applicability,

then the Closed-Eye EnKF must be used, at the cost of a lower heuristic performance.

The forecast during the dry time period D is depicted in Fig. 10c and 10d. In the observed water content range the Closed-Eye EnKF performs better than the Standard EnKF. This is the result that we expected. We were able to estimate the believed true material properties in the observed water content range and can make predictions in this range as long as the basic assumptions

of the Richards equation are met.

Both EnKF can describe the topmost TDR better than the lower TDRs. This is expected as well, since the first TDR measures the strongest dynamics and hence dominates the parameter estimation. The small deviations in the other TDR sensors show, that the assumption of Miller similarity, including the linear interpolation in between is capable to improve the representation. It cannot represent heterogeneity completely, however.




Below the observed soil water content we do not expect predictive capabilities, since the material properties are heuristic and hence cannot be applied outside of the calibrated range. Nevertheless we can see, that the Closed-Eye EnKF does describe the topmost TDR acceptably. Again, we see deviations in the two lower TDR sensors. This time even stronger. Again, we attribute this to the limited description of the heterogeneity with Miller scaling.

As a last comment, we did investigate the possibility of improving the results by estimating a multiplicative factor to the uncertain evaporation in analogy to Wollschläger et al. (2009). This factor can improve the estimation of the topmost TDR towards later times, but does not change the qualitative results above.

## 4    Conclusions

In this study we improved the representation of soil water movement in a 1D soil profile at the Grenzhof test site close to Heidelberg, Germany with an EnKF based on water content measurements by TDR probes.

We assess key uncertainties of this specific representation. These are initial conditions, soil hydraulic parameters, Miller scaling factors (describing small-scale heterogeneity), upper boundary condition, and the local equilibrium assumption by the Richards equation. These were accounted for in the process. Other components deemed to be of lesser importance and were neglected.

The most noteworthy ones of these are hysteresis, violations of the 1D assumption and measurement biases, which might affect our results.

We designed a three stage approach to directly represent and estimate all key uncertainties, except for errors caused by the local hydraulic equilibrium assumption. These intermittent errors are handled by introducing a closed-eye period. Due to an iterative approach we can perform the estimation on a single rain event.

In the first stage, the prior is improved. In our case this is the initial water content distribution and Miller scaling factors. A good prior for the Miller scaling proved particularly beneficial, because during phases without dynamics, water content distributions generated by heterogeneity cannot be distinguished from low hydraulic conductivities.

The second stage is the iterative application of a Standard EnKF with an augmented state to improve the representation. This state consists of soil water contents, soil hydraulic parameters, Miller scaling factors, and the upper boundary condition, to

improve the representation. This approach incorporates possible errors in the dynamics into the augmented state components. Since parameters are assumed to be constant in time, we can detect intermittent errors in the dynamics from fluctuating parameters.

The third stage is the application of a Closed-Eye EnKF, that only estimates the full augmented states when and where we assume the dynamics to be correctly represented. Outside of this range we 'close an eye' and do not estimate the previously

varying parameters. In this way, the state is guided through this ill-represented phase and the full estimation is picked up afterwards again.



The approach showed that homogeneous soil layers are not sufficient to represent heterogeneity. We were forced to also assume small-scale heterogeneity by observations of consistently higher water contents at probes that are located higher up within a soil layer. This emphasizes the role of heterogeneity which must be considered more extensively in studies that rely on local measurements like the used TDR measurements.

We employed a simplified representation of heterogeneity by estimating Miller scaling factors at measurement locations and interpolation between them. This captures the main features. Miller scaling is not capable to represent the heterogeneity completely, however. Predictions for the first layer showed, that measurements from the topmost TDR can be predicted better than the measurements of the second and third TDR probes. We attribute this to limitations of the Miller scaling. The parameter estimation is primarily influenced by the topmost TDR. Describing the other measurements with the same parameter set scaled

by the Miller factor cannot fully describe the material properties there.

An iterative Standard EnKF was used to successfully estimate soil parameters, Miller scaling parameters and upper boundary condition. It was capable to predict the rain event that was used for its calibration very well. However, it fails at the prediction of a different rain event. The reason is the violation of the local equilibrium assumption by the Richards equation (Eq. 1) during

rain events. This can be seen from variations of the parameters that are larger than the change from initial to final value during later iterations. Non-represented errors are incorporated into the parameters. Since these errors are different for different rain events (with different strengths and different local non-equilibrium), they cannot be predicted.

The Closed-Eye EnKF omits these errors in the parameters and yields better predictions during dry periods when the local

equilibrium assumption is fulfilled. As a consequence, however, its predictions are worse during rain events, when the local equilibrium assumption is violated.

During the closed-eye phase, a description of these uncertainties caused by the non-equilibrium is desirable to be able to optimally guide the states through this phase. In our case, the method still performed well without such a description, as the errors were small enough to be compensated by the adaptive covariance inflation.

Our approach is capable to find parameters closer to the believed true material properties of the soil than a standard EnKF. Predictions during rain events would require an additional representation of the dynamics during the rain event, though. Still, our approach shows a way to limit the incorporation of errors into parameters and is one step towards the concept of Knowledge Fusion - the consistent aggregation of all information pertinent to some observed reality.

*Author contributions.* HHB designed, implemented, performed, and analyzed the presented study. SJ provided computational software. DB provided discussions on the statistical foundations from a particle filter point of view. KR came up with the idea of knowledge fusion and the closed-eye period. All authors participated in continuous discussions. HHB prepared the manuscript with contributions from all authors.



*Acknowledgements.* This research is funded by Deutsche Forschungsgemeinschaft (DFG) through project RO 1080/12-1.



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





**Table 1.** Prior uncertainties of the representation of soil water movement at the Grenzhof test site during October and November 2011. We assess the uncertainty caused by assumptions in a qualitative way: small, no need to represent the uncertainty explicitly; intermediate, it might be necessary to represent these uncertainties, we decide not to but keep them in mind; large, uncertainties must be represented with the goal to reduce them.

| Assumption | Uncertainty |
| --- | --- |
| **Dynamics:** | |
| Single phase description (Richards equation) | small |
| Local equilibrium (Richards equation) | small-**large** |
| 1D, no sources and sinks | intermediate |
| **Material Properties:** | |
| Mualem-van Genuchten parameterization for material properties | small |
| Mualem-van Genuchten parameters from Wollschläger et al. (2009) | **large** |
| Mualem-van Genuchten parameters constant in time | small |
| No hysteresis | intermediate |
| Large-scale architecture described with layers | small |
| Small-scale architecture described with Miller scaling | small-intermediate |
| Miller scaling factors are 1 | **large** |
| **Forcing:** | |
| Upper boundary condition from rain gauge and FAO Penman-Monteith eq. | **large** |
| Lower boundary condition constant groundwater table decoupled with soil layer | small |
| Linear interpolation for initial condition | **large** |
| | |
| Numerical representation | small |
| Water content measurements without biases and uncertainty of 0.01 | intermediate |





**Table 2.** Soil hydraulic parameters as estimated by Wollschläger et al. (2009) and our initial uncertainties for the Standard EnKF

| Layer | $\alpha$ [1/m] | $\log_{10}(K_0)$ [m/s] | $n$ | $\theta_s$ | $\theta_r$ | $\tau$ |
|---|---|---|---|---|---|---|
| 1 | 4.8±2.0 | -4.6±1 | 1.61±0.20 | 0.39 | 0.020 | 0.5 |
| 2 | 11.0±2.0 | -5.3±1 | 1.80±0.20 | 0.40 | 0.070 | 0.5 |
| 3 | 7.4±2.0 | -6.0±1 | 1.43±0.20 | 4.27±0.02 | 0.090 | 0.5 |
| 4 | 2.9±2.0 | -5.4±1 | 1.36±0.20 | 4.33±0.02 | 0.100 | 0.5 |
| 5 | 14.5 | -4.1 | 2.68 | 0.43 | 0.045 | 0.5 |

Parameters without uncertainty are not included in the augmented state and are not estimated. The saturated
water content $\theta_s$ of the layers 3 and 4 were already estimated during the prior estimation, because
measurement values exceeded the value of 0.36 estimated by Wollschläger et al. (2009).





**Table 3.** Results for the soil hydraulic parameters estimated with the Standard EnKF.

| Layer | $\alpha$ [1/m] | $\log_{10}(K_0)$ [m/s] | $n$ | $\theta_s$ | $\theta_r$ | $\tau$ |
|---|---|---|---|---|---|---|
| 1 | 4.7±0.3 | -5.23±0.04 | 1.43±0.01 | 0.39 | 0.020 | 0.5 |
| 2 | 10.1±0.7 | -4.08±0.25 | 1.66±0.02 | 0.40 | 0.070 | 0.5 |
| 3 | 8.2±1.0 | -6.04±0.31 | 1.33±0.03 | 0.42±0.01 | 0.090 | 0.5 |
| 4 | 2.3±0.3 | -5.71±0.40 | 1.31±0.01 | 0.43±0.01 | 0.100 | 0.5 |

Parameters without uncertainty are not included in the augmented state and are consequently not estimated.





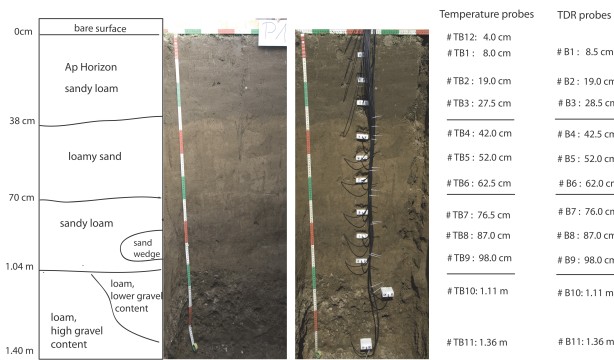

**Figure 1.** Soil profile at the Grenzhof test site close to Heidelberg, Germany (Schenk, 2011). It consists of four different soil layers and is equipped with twelve temperature probes and eleven TDR probes. The surface is kept free from vegetation.



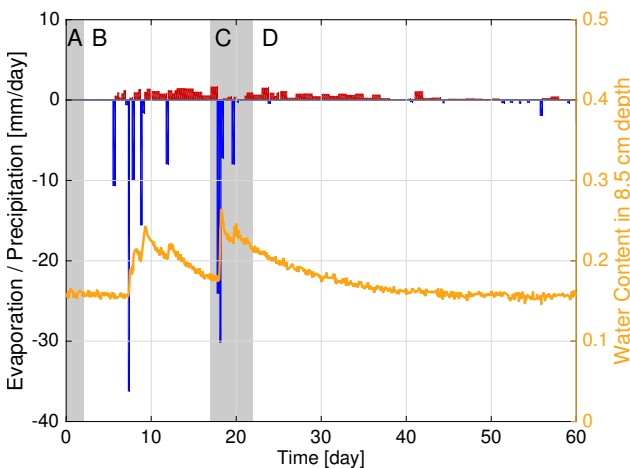

**Figure 2.** Boundary condition and topmost TDR from Oktober 1 (day 1) until November 29 (day 60) at the Grenzhof test site. We distinguish 4 sections: A (day 1-2), B (day 3-17), C (day 18-22) and D (day 23-60). The evaporation is calculated using the reference FAO Penman-Monteith equation. Before the first rain the evaporation is set to 0, due to a previous dry period of over a month.





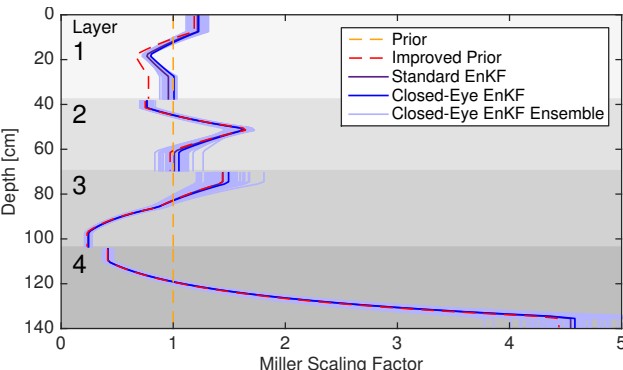

**Figure 3.** Mean values of the Miller scaling field. Soil layers are indicated by different gray scales. The heterogeneity is a priori unknown and the according prior is set to 1. The natural logarithm of the scaling factors is estimated at the measurement locations and interpolated linearly between (with constant extrapolation to layer boundaries). Already the improved prior can describe the main features. The further estimations with the Standard and Closed-Eye EnKF lead to further small changes mainly in the first layer. For the Closed-Eye EnKF the actual ensemble indicating the uncertainties is additionally shown.





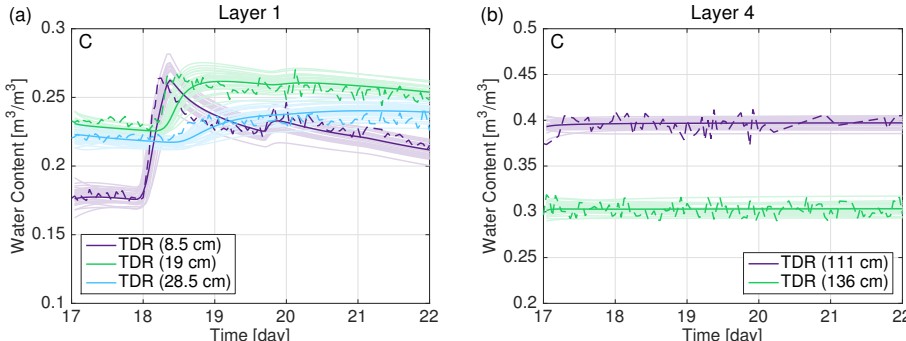

**Figure 4.** Forecast at the measurement positions of the top and bottom layer during time period C. The solid lines show the mean value of a total of 100 different forecasts. The pale colors show the results from 25 of these with soil parameters, Miller scaling parameters and boundary condition sampled from the distributions estimated with the Standard EnKF and the initial condition, that was actually used for the estimation itself. The dashed line shows the measurements.





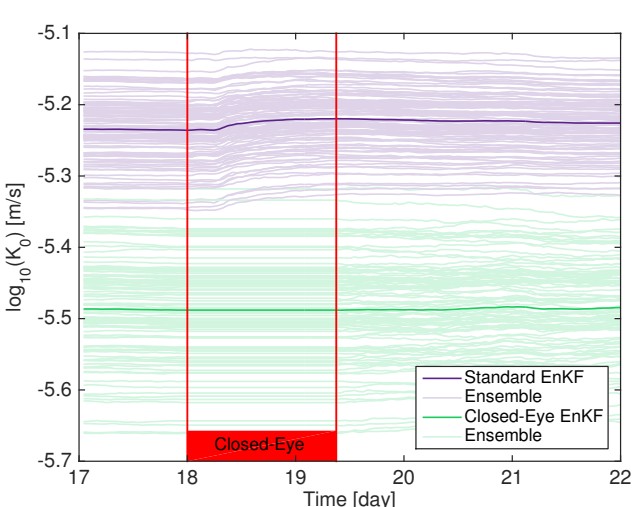

**Figure 5.** Saturated conductivity $K_0$ during the last iteration of the Standard EnKF and the Closed-Eye EnKF.




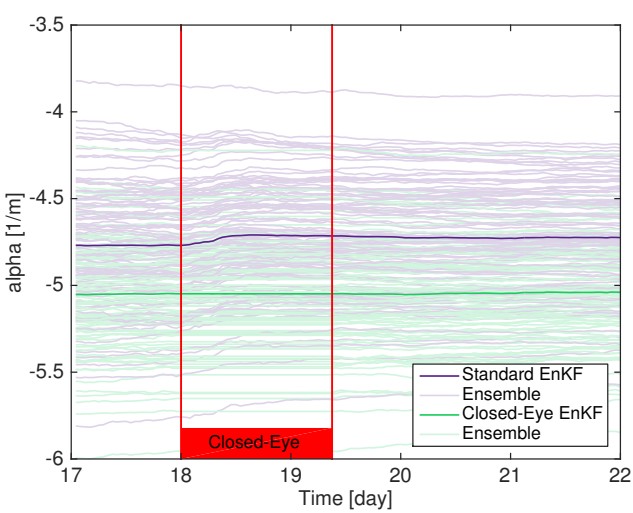

**Figure 6.** Parameter $\alpha$ during the last iteration of the Standard EnKF and the Closed-Eye EnKF.





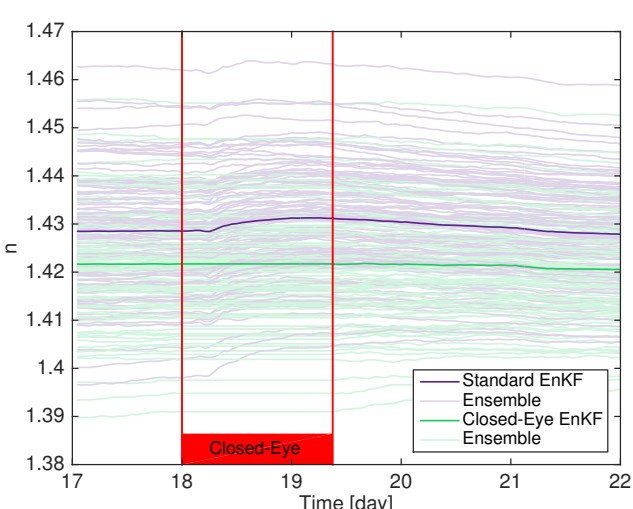

**Figure 7.** Parameter $n$ during the last iteration of the Standard EnKF and the Closed-Eye EnKF.





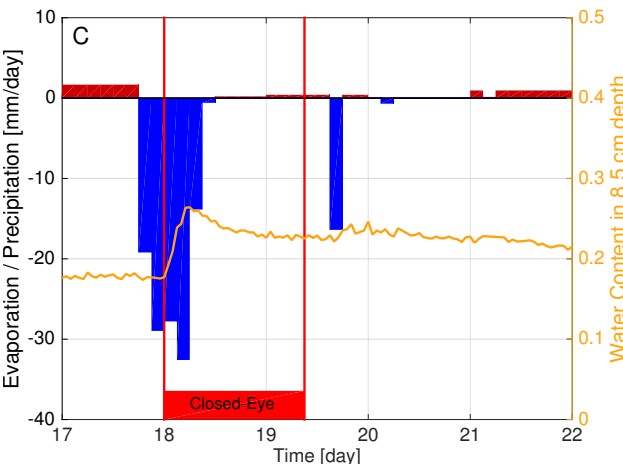

**Figure 8.** Boundary condition along with the topmost TDR water content measurements during time period C. The closed-eye time period was determined based on the changes in parameter $K_0$. During this time the dynamics cannot describe the water content measurements any more, most likely because of the violation of the local equilibrium assumption.





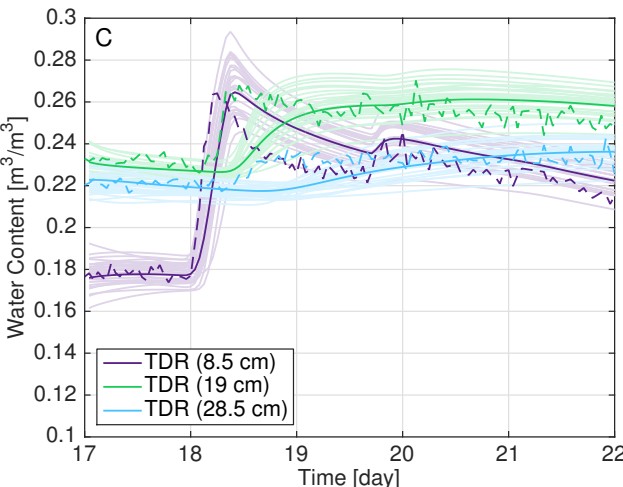

**Figure 9.** Forecast at the measurement positions of the first layer during time period C with the results from the Closed-Eye EnKF. The solid lines show the mean value of a total of 100 different forecasts. The pale colors show the results from 25 of these with soil parameters, Miller scaling parameters and boundary condition sampled from the distributions estimated with the Closed-Eye EnKF and the initial condition, that was actually used for the estimation itself. The dashed line shows the measurements.





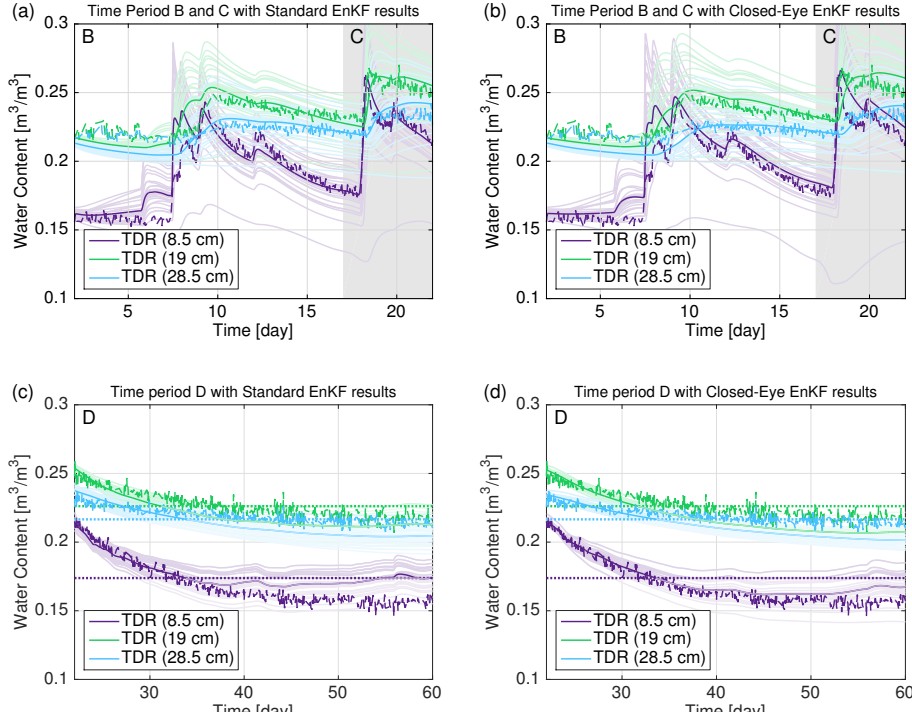

**Figure 10.** Forecast at the measurement positions of the first layer during time period B and C (a,b) as well as D (c,d). The solid lines show the mean value of a total of 100 different forecasts. The pale colors show the results from 25 of these with soil parameters and Miller scaling parameters sampled from the distributions estimated with the Standard EnKF (a,c) and Closed-Eye EnKF (b,d). The dashed line shows the measurements, the doted line the lowest water content measured during time period C, indicating the lower limit of the water content range, for which the parameters are deemed valid.