# Peer review of "EnKF with closed-eye period - towards a consistent aggregation of information in soil hydrology"

_Hydrology and Earth System Sciences, 2016_

## Referee Comment (RC1) · Anonymous Referee #1 · 11 Jul 2016

**Review of „EnKF with closed-eye period – towards a consistent aggregation of information in soil hydrology"**

**Summary:** the manuscript presents a study on parameter estimation for an unsaturated zone model. The authors use real soil water content data to estimate parameters and forcings for a 4 layer model using the EnKF. A full iteration of the filter, divided into different steps is introduced to improve the estimation process. During a rain event it is observed that the estimation gets flawed by the models inability to model preferential flow and the observations corresponding to this rain event are excluded from the parameter estimation. This is called a closed-eye period and it is observed that this improves the stability of the estimated parameters.

**General comments:** The manuscript is interesting and is filling a missing space in the literatur as being a reasonably controlled test case using real data. However, the manuscript is sometimes difficult to overview and the proposed method feels very ad hoc and leaving a number of open questions. Suggestion is major revision.

(throughout the text references to the manuscript are given as: P9,L78 = page 9 Line 78)

**Major concerns**

1. **Data assimilation vs calibration**
   a. The manuscript introduces the concept of a fully iterative filter, which is an approach that I have never seen in the literature before (and obviously the authors neither since there are no citations on this part). Iterative filters are not uncommon, but they are usually local and not global, or restart versions where the model is restarted and ran until next observations. Some relevant references and comparisons to other iterative approaches would be needed as well as a good motivation to the current approach. The authors approach also shows clear similarities to the precalibration exercise by Hubert et al 2010, to the iterative Kalman Ensemble generator by Nowak 2009 (see bottom for references) and to the SODA by Vrugt et al 2005 (already cited, but not compared to the suggested approach).

   b. Further, the suggested approach seems to me a lot more similar to a batch calibration than a filtering, as the same data is used over and over again to calibrate the model towards one final parameter estimate. As it to me is unclear what effect the top boundary estimation has (see below), and as the truly dynamic parts of the data are removed, I have a problem seeing why a filter and not a proper calibration would provide a good solution, given that the filters after all are after all suboptimal for the unsaturated zone?

2. **Top boundary and its update**
   a. The top boundary leaves a few questions unanswered. It is claimed throughout the manuscript that the top boundary is updated and that this is largely also the reason for the use of a data assimilation method rather than a calibration. However, the authors neither show nor discuss the result of these updates, leaving a reader wondering about the necessity of this inclusion.

   b. Further, I do not understand what boundary was used to drive the model. On P8,L10 it is written that the temporal resolution is daily unless a change of mode occurs. On the other hand, all the figures show finer resolved top boundary, and also the scenario descriptions are dealing with temporal resolutions of 10 minutes when describing the rain intensity. If these events are smoothed across a full day (or if the event is shorter than a day, across a full event), it is a considerable smoothing, and it would be much more preferred if the authors also plotted the forcings they are using. Further, wouldn't this risk effecting the parameter estimation during the rain event, if reality has a much stronger peak than the model? Or is the first observation so far down that this has no effect? Please clarify!

3. **Purpose of model?**
   A nice and stable model is calibrated, that can predict water movement as long as nothing really happens. I find the result very interesting, as it properly questions the use of the standard Richards equation for modelling the unsaturated zone, given that the rainfalls that caused the problems here are quite moderate. The authors should take their space to consider the implications of what they show; can we use the Richards equation in the field? What purpose does a model have that is seemingly data driven (P11,L15) and that cannot predict during rainfall?

4. **Memory**
   Since the model used is disconnected from the groundwater, all information within the model has to travel from the top to the bottom in the 1D column. As the lower layers are more dependent on a longer memory to properly assess their behavior (e.g. it takes longer before we see an effect of a rain event in the second layer than in the first), there is a good risk that we smooth this information when continuously updating the first layer.

   a. Wouldn't a continues update of the top layer risk always smoothing the model such that the estimation in the bottom layers becomes difficult?
   b. Could this be helped by the closed-eye period (as here, the rain event that is too strong for the top layer, may give valuable insight in the lower ones and if the model is not updated during this period, the front reaches down)?

5. **Validation**
   Using data that has been used 10-20 times to calibrate a model with as a validation set is not a particularly strong case. More attention could be given to scenario D, where it is nicely shown that the CE-model has a much better performance than the Standard one. Even better would it

be if one of the three observations in the top layer could be taken out of the calibration and used for validating the model; this would be a strong case.

6. **All parameters needed?**
The bottom ¾ of the model is only briefly discussed in the manuscript. That nothing happens in layer 4 is not so surprising, but how does it look for layers 2 and 3? Do the infiltration fronts reach down here and how is it with the parameter estimation? If nothing happens, do we need to estimate them?
How is the effect of the closed-eye period on the second layer?

7. **Overview**
The authors could consider including a nice block diagram to make their approach clearer (which time period is used for what and where is it iterated and what is fed to where?). Two example of unclearness: 1) what is used from 2$^{nd}$ iteration onwards as initial condition for period C; the same output from B despite changed parameters?  2) for the top boundary, is it he finally updated value for each model that is also reused in the next iteration?

8. **How to select a closed-eye period?**
One of the key findings of the manuscript it the improvement using a closed eye period. What is not so clear is how this is to be selected. How can we differentiate between a wanted changes of parameters away from a false prior, from the erroneous updates that the authors show in this work? This feels very ad hoc, and hence also leaves the full paper feeling very problem specific. I think the manuscript would give a more rigorous feeling if this issue was discussed in more depth.

**Minor concerns**

9. **Longer development plots of parameters**
The parameter plots only show the last iteration, however, I would find it highly interesting to know how the development throughout the iterations also looks. Example: mean alpha is initially sampled at 4.8 and has in Figure 6 a value for the standard filter of maybe 4.7 with a jump of 0.05. Hence, it cannot have looked the same during the other 9 iteration, then the value would be different. Similarly, the closed eye filter has a stable value of around 5, but how did the way there from 4.7 look?

10. **Performance of org. model**
Please include in the water content figures, also the performance of the mean of the original model, so that the reader has the possibility to assess the positive development of the model during the filtering.

11. **Why this damping?**
As the selection of a damping parameter is anything but obvious, I think it is useful if authors

using it gives a one sentence motivation to why it was chosen this way!

12. **Resolution of model**
The resolution is uniformed 1cm (P5,L20), which for a model with strong boundary fluxes may not be so small. Has the authors checked that the grid size is not also causing issues? Why is "the effect" minimized by 1 cm and not at 1mm?

13. **Why 100 members?**
The model has 140 unknowns and is 1D, it cannot be a particular difficult to also run a larger ensemble size and reduce the ad hoc tunings, so why is this setup with a small quite small ensemble chosen?

14. **True parameters or heuristic model (was Wollschläger wrong?)**
   a. I'm a bit confused about this discussion. On P12L24 it is stated that the estimated parameters of the closed eye filter better resembles the believed true parameters and that the standard filter ones are more/only heuristic. However, on P14,L1 it is clearly stated that the estimated parameters are all heuristic and only valid in their estimated range, which hence suggests that there are no such things as true parameters.

   b. Also an elaboration on the different result presented here to that of Wollschläger 2009; you get quite different result for a plot quite close by. Are the result so local or where the previously published results not so reliable?

**Technical stuff**
1. Figure 6: negative alpha values?
2. Language: should be checked carefully. E.g. what does the sentence "The forcing or embedding in space is the initial condition" mean?
3. Consider splitting the "Conclusion" section into a "Summary and discussion" (where some of the discussion points taken up in this review would fit) and a short "Conclusion" section which only contains the actual conclusions.

References:

Huber, E., Hendricks-Franssen, H. J., Kaiser, H. P., Stauffer, F., 2011. The Role of Prior Model Calibration on Predictions with Ensemble Kalman Filter. Ground Water 49 (6), 845–858.

Nowak, W., 2009. Best unbiased ensemble linearization and the quasi-linear Kalman ensemble generator. Water Resour. Res. 45 (4)

---

## Referee Comment (RC2) · Anonymous Referee #2 · 15 Aug 2016

In the paper by Bauser et al., the authors perform data assimilation experiments for a 1D real-world case (Grenzhaus site, Germany) for a time period of two months. An iterative EnKF is used to correct states (soil moisture), parameters (layer-based hydraulic conductivity, Miller scaling factors, porosity, van Genuchten parameters) and forcings (upper boundary condition) with measured soil moisture profiles. The authors introduce a so called 'closed-eye' EnKF scheme, in which parameters are not estimated during an intensive rainfall event. This is done in order to prevent the incorporation of model structural errors into the estimated parameters.

General comments:

1) The introduction would benefit from a more detailed description of previous applica-

tions of the EnKF in soil hydrology. The authors should highlight the novelties of their approach compared to previous studies. Additionally, one of the aspects of this paper is a proper description of uncertainties for the utilized test case, so relevant literature on uncertainty description in hydrologic data assimilation should also be cited (e.g. Jafarpour & Tarrahi (2011), Zhang et al. (2015)).

2) More details on the iterative EnKF are needed because this method is not commonly used in hydrologic data assimilation: Are the initial conditions the same for each EnKF iteration? What is the criterion for choosing 50 or 10 EnKF iterations? How do the updated parameters evolve during the different EnKF iterations? In order to judge the EnKF performance better, it would be beneficial if the authors could also provide simulation results with the initial guess of parameters (no data assimilation) as a benchmark.

3) The assimilation time period is rather short and already split into different parts for the estimation of initial conditions/ Miller scaling factors (time periods A/B) and the estimation of the full state-parameter vector (time period C). In between time periods A/B and C there is an inconsistency in terms of ensemble size (40/40 versus 100), the number of EnKF iterations (50/1 versus 10) and an additional perturbation of parameters. I suggest the authors to be more consistent in the data assimilation set-up by using the same meta-parameters (ensemble size/ EnKF iterations) for all time periods. Especially for the change of the ensemble size it is unclear how the states and parameters from the 40 ensemble members are re-sampled to 100 ensemble members (see specific comments below). This data assimilation scheme also does not allow for an equilibration of model states towards the perturbed/ updated parameters. Therefore, I would suggest to add a certain spin-up phase for the ensemble at the beginning of the assimilation phase and between the EnKF iterations. In my opinion, it would also strengthen the paper, if the comparison between standard and 'closed-eye' EnKF could be made for more than one single rainfall event in order to provide more evidence for the effect of model structural errors during rainfall events. This could be achieved,

e.g. by applying parameter estimation with EnKF during both rain events (time period B+C) or by extending the simulation period for additional rain events. It would also be advantageous to use an independent verification period (including one or more rain events, not only the recession phase in time period D) to compare the performance of the updated parameter values from the two assimilation schemes.

4) It is quite unclear from the manuscript how the upper boundary condition is updated. As a filtering method, the EnKF only adapts the state-parameter vector for the current model time step. How is the updated upper boundary condition then incorporated in the subsequent model integration? Additionally, the results for the update of the upper boundary condition are not really discussed in detail. So it is quite unclear, how this update affects the model simulations. The state vector includes almost all components of the water balance for the 1D soil profile. These components are all adapted with the same measurements and it is unclear whether the soil moisture data contain enough information to update all water balance components at once. As the main focus of the paper is on parameter estimation, I suggest the authors to leave out the estimation of the upper boundary condition and only use perturbed forcing data instead.

5) A discussion is missing on how a 'closed-eye' EnKF period can be defined in other practical applications. In the paper, a quite heuristic criterion (maximal parameter change within a defined time period) is used which relies on a prior application of the standard EnKF assimilation scheme. Can such a criterion be generalized? The EnKF is usually applied as a pure sequential method without iterations (often in the context of real-time simulations), so the question is whether the 'closed-eye' period can also be defined properly for such cases, or if the iterative EnKF is a prerequisite for the 'closed-eye' EnKF. An additional question is how such a 'closed-eye' period would be defined for a 3D model where precipitation is spatially distributed and the computational demand is much larger than for a 1D soil column.

6) To my knowledge, the covariance inflation method proposed in Anderson (2009) has not been applied in hydrologic data assimilation so far. It would be interesting to see,

how this inflation scheme affects model results, e.g. by providing some information on the calculated inflation factors or by comparison with a standard EnKF assimilation scheme.

Specific comments:

page 1 line 6: I think the definition of the 'closed-eye EnKF' should already been given here or at least in the introduction.

page 2 line 15-16: This not fully representative for the study. You should also add your experiments with the standard/ 'closed-eye' EnKF here.

page 3 line 22-25: Please clarify what you define as 'non-equilibrium conditions' in this context. Do you mean preferential flow?

page 4 line 20-27: The definition of Miller scaling requires more details. How exactly are equations 5+6 incorporated into your model simulations? I guess you define a scaling factor for each model grid cell and use the parameters of the respective soil layer as the "reference material property". Additionally, why does a scaling factor of 1 introduce large uncertainties? This is not obvious from equations 5+6.

page 5 line 5-9: The TDR measurements for 111 and 136 cm in Figure 4b suggest that there is some dynamics in the lower part of the soil profile. On the contrary, the simulated soil moisture contents are almost constant. This is an indication that there might be some problems with the assigned lower boundary condition. The constant simulated soil moisture values are probably an effect of the sand layer that was introduced as layer 5 which acts as a hydraulic barrier to the upper soil profile. You could try to use the soil hydraulic parameters from layer 4 also for layer 5 to see, if the simulated soil moisture dynamics can be improved in the lower part of the soil profile. Additionally, it is not clear why you have chosen a water table depth of 4m. Is this value based on water level data from surrounding wells?

page 7 line 25: If the soil moisture measurements exceed the calibrated saturated

water, why don't you use the data from the volumetric soil samples (page 5 line 24) as an initial guess?

page 8 line 9-11: Does this mean that you only use daily averages of precipitation? In the data assimilation experiments rather short time periods are investigated and the simulations are compared to hourly soil moisture measurements. So it would make much more sense to also use atmospheric forcing data with a similar time resolution which should be available from the weather station (page 3 line 3-4).

page 8 line 30-31: Which value for the cut-off radius did you use in the 5th order polynomial for soil water content?

page 9 line 2-4: Why is the localization different for hydraulic conductivity and Miller scaling factors? Miller scaling factors are strongly related to hydraulic conductivities and should therefore also have a similar localization.

page 9 line 9-11: Did you also use localization in the derivation of the inflation factors? This is necessary because otherwise uncertainties are increased in model areas that are not updated by the EnKF.

page 10 line 20-23: How are initial Miller scaling factors perturbed?

page 11 line 7-11: It is unclear how the initial conditions for time period C were exactly created. How were the final (updated) states from time period B re-sampled from 40 to 100 ensemble members? Did you perturb the ensemble mean of Miller scaling factors from time period A or each of the 40 ensemble members individually? Was this perturbation constant in space or was the perturbation on the grid cell/ soil horizon level? Please also take into account that this perturbation creates inconsistencies between states and parameters which could be alleviated by a spin-up period.

page 11 line 32 - page 12 line 4: It is not obvious from Figure 5 whether hydraulic conductivity reaches its initial value by the end of time period C. Please be more quantitative here: How large is the change within the 'closed-eye' period and how large is

the difference between initial and final values? It would also be good to show how the update of hydraulic conductivity evolves over the different iterations of the EnKF. Is there a continuous increase of this parameter during the iterations? Additionally, the increase of hydraulic conductivity during the closed-eye period could also be related to the adaptive covariance inflation. The calculation of the inflation factors is influenced by the mismatch between simulations and measurements. As this mismatch increases during the infiltration event, this could also lead to higher inflation factors which increase the Kalman gain and thus the parameter update for this period. Did you check how the inflation factors evolve over this time period? Maybe it's also worthwhile to repeat these simulations without covariance inflation in order to exclude possible artefacts from the inflation scheme.

page 12 line 14-17: The 'closed-eye' EnKF experiments are added on top of the standard EnKF experiments with an additional parameter perturbation. This makes it quite difficult to compare these two experiments. I suggest to use exactly the same model set-up for both experiments, i.e. to repeat the 'closed-eye' EnKF experiment with the same initial conditions as the experiment with standard EnKF. Otherwise, a direct comparison of both methods is not possible. An additional question is why the 'closed-eye' EnKF and the parameter perturbation is only performed for the first soil layer.

page 12 line 23-24: What are the 'believed true material properties'? Given the fact that this is a real-world experiments, where the true material properties are unknown, this is quite speculative.

page 13 line 13-22: Please be more quantitative here by providing performance measures such as root mean square error or Nash-Sutcliffe efficiency.

page 13 line 26-30: In Figure 10b the simulated ensemble mean for each layer is always above the measured soil moisture values at the end of time period C. Why does this offset not appear in the beginning of time period D in Figure 10d? Shouldn't period D start with the simulation results from period C?

page 14 line 28-31: You should also discuss how such a 'closed-eye' period can be defined in practical applications.

Figure 2: Why is there no evaporation flux in time period A and at the beginning of time period B? If is is a dry period, there should also be evaporation.

Technical corrections:

page 13 line 3: Change 'andit' to 'and it'.

Table 2: Saturated soil moisture content for layers 3+4 is above 1. Please correct.

References:

D. Zhang, H. Madsen, M. E. Ridler, J. C. Refsgaard, K. H. Jensen (2015), Impact of uncertainty description on assimilating hydraulic head in the MIKE SHE distributed hydrological model, Advances in Water Resources, 86, 400-413, doi:10.1016/j.advwatres.2015.07.018.

B. Jafarpour, M. Tarrahi (2011), Assessing the performance of the ensemble Kalman filter for subsurface flow data integration under variogram uncertainty, Water Resources Research, 47, W05537, doi:10.1029/2010WR009090.

---

## Author Comment (AC1) · 13 Sep 2016

**Summary:** *the manuscript presents a study on parameter estimation for an unsaturated zone model. The authors use real soil water content data to estimate parameters and forcings for a 4 layer model using the EnKF. A full iteration of the filter, divided into different steps is introduced to improve the estimation process. During a rain event it is observed that the estimation gets flawed by the models inability to model preferential flow and the observations corresponding to this rain event are excluded from the parameter estimation. This is called a closed-eye period and it is observed that this improves the stability of the estimated parameters.*
**General comments:** *The manuscript is interesting and is filling a missing space in*

[Figure]

*the literatur as being a reasonably controlled test case using real data. However, the manuscript is sometimes difficult to overview and the proposed method feels very ad hoc and leaving a number of open questions. Suggestion is major revision.*
*(throughout the text references to the manuscript are given as: P9,L78 = page 9 Line 78)*

**Reply:** We thank the reviewer for the constructive comments and suggestions. We have revised our manuscript (see supplement) taking them into account. For consistency with your references, we will still refer to the unrevised manuscript as well.

**Major concerns**
*1. Data assimilation vs calibration*
*a. The manuscript introduces the concept of a fully iterative filter, which is an approach that I have never seen in the literature before (and obviously the authors neither since there are no citations on this part). Iterative filters are not uncommon, but they are usually local and not global, or restart versions where the model is restarted and ran until next observations. Some relevant references and comparisons to other iterative approaches would be needed as well as a good motivation to the current approach. The authors approach also shows clear similarities to the precalibration exercise by Hubert et al 2010, to the iterative Kalman Ensemble generator by Nowak 2009 (see bottom for references) and to the SODA by Vrugt et al 2005 (already cited, but not compared to the suggested approach).*

**Reply:** Yes, we have not seen the fully iterative approach in the literature as well. We improved the introduction of the approach (page 9, line 12-15) to: "We iterate the whole EnKF scheme and start the next iteration with the final estimation of soil parameters, Miller scaling factors and upper boundary condition of the previous iteration. These iterations differ from the typically applied iterative EnKFs like the Restart EnKF or Confirming EnKF (e.g. Song et al., 2014) and rather resemble the iterations in the Kalman ensemble generator by Nowak (2009), who used a modified EnKF to estimate parameters only. The full iterations applied are required to estimate

constant augmented state components even with a small damping factor and a rather short time period of data. This is especially important for the upper boundary condition, where the time periods of constant values are short."

We have not been aware of the works by Huber et al. (2011) and Nowak (2009). Thank you, for pointing them out to us. We have incorporated Nowak (2009) in the text shown above (page 9, line 12-15) and Huber et al. (2011) in the description of the prior estimation (page 10, line 14): "Highly uncertain properties can exacerbate the performance of the EnKF. For example Huber et al. (2011) showed that the state estimation with an EnKF is superior in case of properly precalibrated model parameters. In our case, there is no prior knowledge about small scale heterogeneity, which can lead to difficulties in the estimation process:"

The SODA approach by Vrugt et al. (2005) is comparable to the Standard EnKF approach in our manuscript, since he as well estimated parameters, while continuously updating the states with an EnKF. But, the estimation with the augmented state instead of the shuffled complex evolution metropolis algorithm is more computational efficient. We added this in the description of the Standard EnKF (page 11, line 6).

*b. Further, the suggested approach seems to me a lot more similar to a batch calibration than a filtering, as the same data is used over and over again to calibrate the model towards one final parameter estimate. As it to me is unclear what effect the top boundary estimation has (see below), and as the truly dynamic parts of the data are removed, I have a problem seeing why a filter and not a proper calibration would provide a good solution, given that the filters after all are after all suboptimal for the unsaturated zone?*

**Reply:** Yes, our approach is not a typical filter application for state estimation. Similar to a batch calibration we focus on the parameter estimation, but we argue that our approach goes beyond. Unlike the batch calibration, the EnKF continuously updates the state and could incorporate model errors. Due to the continuous parameter estimation we were able to detect the times when the model assumptions do not

hold (the closed-eye period). Due to the state estimation we can remove these times (which in this case correspond to the dynamic parts of the data). The state estimation enables us to guide the state through this time with a erroneous model and allows us to pick up the estimation afterwards again.

The estimation of the boundary condition is required to prevent the introduction to biases into the parameters. This again is required for a proper detection of the closed-eye period (see below).

**2. Top boundary and its update**

*a. The top boundary leaves a few questions unanswered. It is claimed throughout the manuscript that the top boundary is updated and that this is largely also the reason for the use of a data assimilation method rather than a calibration. However, the authors neither show nor discuss the result of these updates, leaving a reader wondering about the necessity of this inclusion.*

*b. Further, I do not understand what boundary was used to drive the model. On P8,L10 it is written that the temporal resolution is daily unless a change of mode occurs. On the other hand, all the figures show finer resolved top boundary, and also the scenario descriptions are dealing with temporal resolutions of 10 minutes when describing the rain intensity. If these events are smoothed across a full day (or if the event is shorter than a day, across a full event), it is a considerable smoothing, and it would be much more preferred if the authors also plotted the forcings they are using. Further, wouldn't this risk effecting the parameter estimation during the rain event, if reality has a much stronger peak than the model? Or is the first observation so far down that this has no effect? Please clarify!*

**Reply:** We agree, that we did not discuss the estimation of the upper boundary in sufficient detail in the manuscript. An estimation of the upper boundary condition is not mandatory. A stochastic representation of the uncertainties could be sufficient. But we argue, that a bias in the upper boundary condition could induce a bias in the estimated parameters. Especially if this bias mainly occurs during the rain event, it

could cause a parameter shift during the rain event, similar to the one observed during the closed-eye period. To prevent this, we do estimate the boundary condition. This requires the decreased temporal resolution, which could lead to short-term parameter variations within the rain event. We think that this downside is subordinate to the removal of the bias, especially since it is strongly alleviated by the disspiative soil water movement.

We improved the description of the boundary condition in the manuscript (page 8, line 5-11): "The expansion to an augmented state changes the propagation in time. Each component needs an individual forward propagation. We assume the soil hydraulic parameters and Miller scaling factors to be constant in time. This is not possible for the upper boundary condition where the forward equation is unknown from a soils perspective. However, measurements are available to estimate the evaporation and precipitation. Hence, we assume the forward model constant until a new estimation is available. Then, we switch to the estimated boundary condition. To base the improvement of the upper boundary condition on several measurements, we reduce the temporal resolution of precipitation measurements to change daily and at transitions between precipitation and evaporation. This means, that the upper boundary condition is treated like the parameters, except that the value can change in the forward propagation. The original temporal resolution of the precipitation data (10 minutes) is not required due to the dissipative nature of the Richards equation, which smooths the infiltration front before it reaches the first TDR sensor in a depth of 8.5 cm. The estimation of the averaged boundary condition will ensure that there is no global bias on the parameter estimation during the rain event, but could lead to small short-time parameter drifts within a rain event." and (page 11, line 34): "As we added the boundary condition into our estimation, we can exclude effects due to a bias in the precipitation to cause this parameter shift. The reduced temporal resolution in the boundary condition could only cause short-term parameter changes within the rain event."

We also show the boundary condition with the applied reduced temporal resolution in

Fig. 8 now.

**3. Purpose of model?**

*A nice and stable model is calibrated, that can predict water movement as long as nothing really happens. I find the result very interesting, as it properly questions the use of the standard Richards equation for modelling the unsaturated zone, given that the rainfalls that caused the problems here are quite moderate. The authors should take their space to consider the implications of what they show; can we use the Richards equation in the field? What purpose does a model have that is seemingly data driven (P11,L15) and that cannot predict during rainfall?*

**Reply:** Yes, we agree that the Richards equation cannot describe preferential flow, which limits it's applicability. Quite to the opposite preferential flow is mostly relevant for the top layers only and less for the lower layers. Consequently the Richards equation is valid there and can be applied.

In our manuscript we are interested in the actual processes. We separated the times with and without preferential flow and show that the Richards equation can really describe the times without preferential flow.

Furthermore, if one is not interested in the processes, but an optimal prediction a batch calibration can yield better results, because there effects of preferential flow are incorporated into the parameters, which then in part compensate the non-represented processes.

**4. Memory**

*Since the model used is disconnected from the groundwater, all information within the model has to travel from the top to the bottom in the 1D column. As the lower layers are more dependent on a longer memory to properly assess their behavior (e.g. it takes longer before we see an effect of a rain event in the second layer than in the first), there is a good risk that we smooth this information when continuously updating*

the first layer.

*a. Wouldn't a continues update of the top layer risk always smoothing the model such that the estimation in the bottom layers becomes difficult?*

*b. Could this be helped by the closed-eye period (as here, the rain event that is too strong for the top layer, may give valuable insight in the lower ones and if the model is not updated during this period, the front reaches down)?*

**Reply:** For the estimation of the second layer, the state of the first layer at the border to the second layer has to be represented as well as possible. We can picture the first layer as a boundary condition for the second layer. This means, that the state estimation of the first layer will improve the parameter estimation of the second layer. Especially during the rain event this state estimation becomes very important. Without it, the infiltration front would reach the second layer at the wrong time, leading to biased parameters there.

The closed-eye period in the first layer does not improve the parameter estimation in the second layer, because the parameters in the first layer are worse at describing the rain event This requires stronger corrections in the state, which actually might make the parameter estimation in the second layer more difficult.

But, we agree, that in case of a proper representation of the preferential flow a better estimation of the second layer might be possible with a longer memory.

**5. Validation**

*Using data that has been used 10-20 times to calibrate a model with as a validation set is not a particularly strong case. More attention could be given to scenario D, where it is nicely shown that the CE-model has a much better performance than the Standard one. Even better would it be if one of the three observations in the top layer could be taken out of the calibration and used for validating the model; this would be a strong case.*

**Reply:** Yes, we agree, taking one sensor out of the calibration and using it for validation would be a strong case. Unfortunately this is impossible in the given scenario. Due to

the detected small scale heterogeneity, this heterogeneity has to be estimated at each sensor location. Without it, it is impossible to predict the water content at this location. This is a fundamental challenge in a heterogeneous medium and the only way to verify the results is the presented internal consistency.

**6. All parameters needed?**
*The bottom $\frac{3}{4}$ of the model is only briefly discussed in the manuscript. That nothing happens in layer 4 is not so surprising, but how does it look for layers 2 and 3? Do the infiltration fronts reach down here and how is it with the parameter estimation? If nothing happens, do we need to estimate them? How is the effect of the closed-eye period on the second layer?*
**Reply:** We have updated Fig. 4 and included the layers 2 and 3. In both layers there is basically no dynamics as well. We agree, that the estimation of these parameters is operationally not required in this case, but we wanted to address the challenge to estimate parameters there. We could see, that we can already estimate heterogeneity and reduce uncertainty in soil parameters. Additionally, we gained information about the robustness of the procedure. The estimated parameters did not reach values, that would be excluded by expert knowledge. Without the representation of heterogeneity the hydraulic conductivity in those layers dropped rapidly to very small values, which could be excluded.

**7. Overview**
*The authors could consider including a nice block diagram to make their approach clearer (which time period is used for what and where is it iterated and what is feeded to where?). Two example of unclearness: 1) what is used from 2nd iteration onwards as initial condition for period C; the same output from B despite changed parameters? 2) for the top boundary, is it he finally updated value for each model that is also reused in the next iteration?*

**Reply:** We followed your advice and included a block diagram. To the specific questions: 1) the same initial condition generated once is used for all iterations. 2) the updated mean value and standard deviation for each time interval are used to generate the boundary condition ensemble for the next iteration. Both is mentioned explicitly in the text now.

**8. How to select a closed-eye period?**
*One of the key findings of the manuscript it the improvement using a closed eye period. What is not so clear is how this is to be selected. How can we differentiate between a wanted changes of parameters away from a false prior, from the erroneous updates that the authors show in this work? This feels very ad hoc, and hence also leaves the full paper feeling very problem specific. I think the manuscript would give a more rigorous feeling if this issue was discussed in more depth.*

**Reply:** The idea of the closed-eye period is not ad hoc and actually generally applicable. In our case we can detected the closed eye-period due to the iterative approach and thus we can determine, based on the assumption of constant parameters, the closed-eye period due to the change in the direction of the parameter update.

We enhanced the discussion accordingly (page 15, line 14-20): "Due to the iterative approach, we could detect the times when the local-equilibrium assumption is violated: the variations of the parameters are larger than the change from initial to final value during later iterations. The changes in the direction of the parameter update then determine the closed-eye period.

Generally, the closed-eye period can be detected, if the operational limits of the model are known. In our case, we base this on the changing parameters, but e.g. a direct detection in the state or forcing could be possible as well.

The Closed-Eye EnKF omits the incorporation of the model structural errors in the parameters and is a generally applicable concept. In this study, it yields better predictions during periods when the underlying assumptions are fulfilled: the drying period after a rain event when there is local equilibrium, showing the strength of the

Richards equation there."

**Minor concerns**
*9. Longer development plots of parameters*
*The parameter plots only show the last iteration, however, I would find it highly inter-esting to know how the development throughout the iterations also looks. Example: mean alpha is initially sampled at 4.8 and has in Figure 6 a value for the standard filter of maybe 4.7 with a jump of 0.05. Hence, it cannot have looked the same during the other 9 iteration, then the value would be different. Similarly, the closed eye filter has a stable value of around 5, but how did the way there from 4.7 look?*
**Reply:** We have added the evolution of the parameters during all iterations. Especially during the first iteration parameters can overshoot to compensate other parameters, leading to the observed behaviour. Additionally, the mean value is not perfectly conserved from one iteration to the next, because the ensemble is not kept, but sampled new from a Gaussian distribution with mean and standard deviation of the final parameters of the previous iteration.

*10. Performance of org. model*
*Please include in the water content figures, also the performance of the mean of the original model, so that the reader has the possibility to assess the positive development of the model during the filtering.*
**Reply:** We included the original model in Fig. 4. It assumes homogeneous soil layers and cannot describe the observed water contents.

*11. Why this damping?*
*As the selection of a damping parameter is anything but obvious, I think it is useful if authors using it gives a one sentence motivation to why it was chosen this way!*
**Reply:** We based our choice on values seen in the literature. We enhanced the
text on page 8, line 20-21: "Hendricks Franssen and Kinzelbach (2008) investigated damping factors between 0.1 and 1.0, while Erdal et al. (2014) employed a damping factor of 0.3 for the parameters. The smaller the value, the better can a non-linearity be handled, but the slower will the estimated value approach the optimal value. We choose the values d = [1.0, 0.1, 0.1, 0.1]. This means, that based on the water content measurements, there is no damping for the update of the water content state, but a strong damping for soil hydraulic parameters, Miller scaling factors and upper boundary condition."

**12. Resolution of model**
*The resolution is uniformed 1cm (P5,L20), which for a model with strong boundary fluxes may not be so small. Has the authors checked that the grid size is not also causing issues? Why is "the effect" minimized by 1 cm and not at 1mm?*
**Reply:** We improved our wording to: "We choose a resolution of 1 cm. For the represented soil and boundary conditions this resolution is sufficient to resolve the occurring fronts smoothly, so that we can neglect effects caused by the numerical solver."

**13. Why 100 members?**
*The model has 140 unknowns and is 1D, it cannot be a particular difficult to also run a larger ensemble size and reduce the ad hoc tunings, so why is this setup with a small quite small ensemble chosen?*
**Reply:** We did not investigate the ensemble size, as 100 members proofed to be sufficient. By increasing the ensemble size we only see the localization as a possibility to reduce the applied improvements. We do not see the localization as an ad hoc tuning, though. A limited information range is physically based.
The damping factor is included to attenuate the effect of linearizing non-linear relations, the covariance inflation is added to prevent filter inbreeding, which can be caused

non-Gaussian distributions and the prior estimation of Miller scaling factors is added due to otherwise occurring wrong correlations between hydraulic conductivity and unrepresented heterogeneity.

**14. True parameters or heuristic model (was Wollschläger wrong?)**

*a. I'm a bit confused about this discussion. On P12L24 it is stated that the estimated parameters of the closed eye filter better resembles the believed true parameters and that the standard filter ones are more/only heuristic. However, on P14,L1 it is clearly stated that the estimated parameters are all heuristic and only valid in their estimated range, which hence suggests that there are no such things as true parameters.*
*b. Also an elaboration on the different result presented here to that of Wollschläger 2009; you get quite different result for a plot quite close by. Are the result so local or where the previously published results not so reliable?*

**Reply:** We agree, there are no true parameters, but there are, what we call 'believed true material properties', which are only valid in the observed range. We have added our definition (page 12, line 24): "This means, if the parameters are constant in time, we believe, that the parameters can represent reality in the observed water content range during times, when the underlying assumptions hold."

Wollschläger et al. (2009) were not wrong. The used data comprised only one TDR sensor in each layer. The obtained results are based on the assumption of homogeneous layers and are optimal in the sense of an inversion when a true model is assumed. As shown in the manuscript the representation of heterogeneity is important, though.

**Technical stuff**

*1. Figure 6: negative alpha values?*

**Reply:** The sign of the $\alpha$ values is connected to the choice how the potential is defined. If bound water content states are chosen to have a negative potential, the $\alpha$

values have to be negative.

*2. Language: should be checked carefully. E.g. what does the sentence "The forcing or embedding in space is the initial condition" mean?*

**Reply:** We corrected the sentence and improved formulations throughout the manuscript.

*3. Consider splitting the "Conclusion" section into a "Summary and discussion" (where some of the discussion points taken up in this review would fit) and a short "Conclusion" section which only contains the actual conclusions.*

**Reply:** We kept the existing sections, but improved the Conclusion.

Please also note the supplement to this comment:
http://www.hydrol-earth-syst-sci-discuss.net/hess-2016-296/hess-2016-296-AC1-supplement.pdf

**Supplement:**

[revised manuscript text omitted]

---

## Author Comment (AC2) · 13 Sep 2016

*In the paper by Bauser et al., the authors perform data assimilation experiments for a 1D real-world case (Grenzhaus site, Germany) for a time period of two months. An iterative EnKF is used to correct states (soil moisture), parameters (layer-based hydraulic conductivity, Miller scaling factors, porosity, van Genuchten parameters) and forcings (upper boundary condition) with measured soil moisture profiles. The authors introduce a so called 'closed-eye' EnKF scheme, in which parameters are not estimated during an intensive rainfall event. This is done in order to prevent the incorporation of model structural errors into the estimated parameters.*

We thank the reviewer for the constructive comments and suggestions. We have revised our manuscript (see supplement) taking them into account. For consistency with your references, we will still refer to the unrevised manuscript as well.

**General comments:**

*1) The introduction would benefit from a more detailed description of previous applications of the EnKF in soil hydrology. The authors should highlight the novelties of their approach compared to previous studies. Additionally, one of the aspects of this paper is a proper description of uncertainties for the utilized test case, so relevant literature on uncertainty description in hydrologic data assimilation should also be cited (e.g. Jafarpour & Tarrahi (2011), Zhang et al. (2015)).*

**Reply:** We have extended our introduction. We had not been aware of the works by Jafarpour and Tarrahi (2011) and Zhang et al. (2015) and cite them now.

We also added our introduced approach into the introduction (starting page 2, line 10): "Synthetic cases offer the advantage of a direct control and knowledge about all uncertainties, which is a challenge in real world cases. The characterization of uncertainties is critical for the success of a data assimilation scheme (Liu et al., 2012). Assuming the wrong uncertainty sources can lead to poorer performance (e.g. Crow and Van Loon, 2006; Zhang et al., 2015). In case the correct uncertainty is unknown it can be better to overestimate uncertainties than to disregard or underestimate them (e.g. Jafarpour and Tarrahi, 2011; Zhang et al., 2015).

In this study, we focus on a real world case to address the challenge of consistent aggregation of the information. We exercise this aggregation with the EnKF on a small situation: a 1D soil profile equipped with time domain reflectometry (TDR) probes measuring water content, during a time period of less than 2 months. We assess all uncertainties in the representation of this particular situation qualitatively and design a three stage approach to reduce the largest uncertainties or to consider them appropriately: First, we improve the prior knowledge (initial condition and small
scale heterogeneity) to facilitate the subsequent estimation. Second, we perform an assimilation with a standard EnKF approach with an augmented state to directly reduce uncertainties in states, soil hydraulic parameters, small scale heterogeneity and upper boundary condition. We introduce iterations over the complete EnKF scheme to cope with the short data set, and to determine times when the underlying local equilibrium assumptions by the Richards equation are violated. We define this specific time as the closed-eye period, because in the third stage, we only estimates states, but keep parameters constant during this time and thus prevent the incorporation of the uncertainties in the dynamics into the parameters. The estimation of soil hydraulic parameters is only performed before and after this closed-eye period.

The remainder of the manuscript is organized as follows: Section 2 describes our understanding of (i) a representation of a physical system in general, (ii) the hydrological system we want to represent, (iii) the specific representation and its uncertainties in detail, (iv) the concept of knowledge fusion and (v) the EnKF. Section 3 presents the three stages: (i) Improving the prior, (ii) Standard EnKF and (iii) Closed-Eye EnKF and compares the prediction capabilities of Standard and Closed-Eye EnKF. In section 4 we present our conclusion and their implications."

*2) More details on the iterative EnKF are needed because this method is not commonly used in hydrologic data assimilation: Are the initial conditions the same for each EnKF iteration? What is the criterion for choosing 50 or 10 EnKF iterations? How do the updated parameters evolve during the different EnKF iterations? In order to judge the EnKF performance better, it would be beneficial if the authors could also provide simulation results with the initial guess of parameters (no data assimilation) as a benchmark.*
**Reply:** We improved the introduction of the iterative approach (page 9, line 12-15):"We iterate the whole EnKF scheme and start the next iteration with the final estimation of soil parameters, Miller scaling factors and upper boundary condition of the previous iteration. These iterations differ from the typically applied iterative EnKFs like the

[Figure]

Restart EnKF or Confirming EnKF (e.g. Song et al., 2014) and rather resemble the iterations in the Kalman ensemble generator by Nowak (2009), who used a modified EnKF to estimate parameters only. The full iterations applied are required to estimate constant augmented state components even with a small damping factor and a rather short time period of data. This is especially important for the upper boundary condition, where the time periods of constant values are short."

The initial conditions are kept the same for each iteration. We added this information to the manuscript. We followed the advice of Referee #1 and include a block diagram to better illustrate the three stage approach. We added the evolution of the parameters during all iterations and show the results of the initial guess parameters together with the results of the Standard EnKF.

*3) The assimilation time period is rather short and already split into different parts for the estimation of initial conditions/ Miller scaling factors (time periods A/B) and the estimation of the full state-parameter vector (time period C). In between time periods A/B and C there is an inconsistency in terms of ensemble size (40/40 versus 100), the number of EnKF iterations (50/1 versus 10) and an additional perturbation of parameters. I suggest the authors to be more consistent in the data assimilation set-up by using the same meta-parameters (ensemble size/ EnKF iterations) for all time periods. Especially for the change of the ensemble size it is unclear how the states and parameters from the 40 ensemble members are resampled to 100 ensemble members (see specific comments below). This data assimilation scheme also does not allow for an equilibration of model states towards the perturbed/ updated parameters. Therefore, I would suggest to add a certain spin-up phase for the ensemble at the beginning of the assimilation phase and between the EnKF iterations. In my opinion, it would also strengthen the paper, if the comparison between standard and 'closed-eye' EnKF could be made for more than one single rainfall event in order to provide more evidence for the effect of model structural errors during rainfall events. This could be achieved, e.g. by applying parameter estimation with EnKF during both rain events*

*(time period B+C) or by extending the simulation period for additional rain events. It would also be advantageous to use an independent verification period (including one or more rain events, not only the recession phase in time period D) to compare the performance of the updated parameter values from the two assimilation schemes.*

**Reply:** The number of iterations between time period A and C was chosen differently, due to the different assimilation time (2 days and 5 days). The ensemble size was increased from time period A to C, because the number of estimated parameters was increased as well. The ensemble size in time period B was chosen as 100 as well, so that we can directly use the states as initial condition, which we now clarify in the text. The information about parameters is passed as mean value and standard deviation, which allows the change in the ensemble size.

We agree that a spin-up phase is conceptually correct. We see the state estimation time period B as an initial spin-up phase to achieve a good estimate of the initial condition. A spin-up phase without state estimation is not suitable due to the unrepresented structural model errors, which could lead to a biased initial condition.

We investigated the benefit of incorporating this initial spin-up into the iterations. The spin-up lead to stronger correlations between states and parameters in the beginning. Without the spin-up the initial correlations are small, but develop rapidly. We see this as an internal spin-up phase. This might require more iterations, but compared to the additional computational time by the spin-up phase (time period B is 3 times longer than time period C, which then quadruples the computation time), these costs are small. Because of that we decided not to include a spin-up phase into the iterations.

We agree, that the concept of the closed-eye period should be investigated on further rain events and verification periods. This will be part of our future work. In this manuscript we focus on putting forward the concept.

*4) It is quite unclear from the manuscript how the upper boundary condition is updated. As a filtering method, the EnKF only adapts the state-parameter vector for the current model time step. How is the updated upper boundary condition then incorporated in*

*the subsequent model integration? Additionally, the results for the update of the upper boundary condition are not really discussed in detail. So it is quite unclear, how this update affects the model simulations. The state vector includes almost all components of the water balance for the 1D soil profile. These components are all adapted with the same measurements and it is unclear whether the soil moisture data contain enough information to update all water balance components at once. As the main focus of the paper is on parameter estimation, I suggest the authors to leave out the estimation of the upper boundary condition and only use perturbed forcing data instead.*

**Reply:** The upper boundary condition is part of the augmented state and is treated like a parameter, except that the value can change in the forward propagation. An estimation of the upper boundary condition is not mandatory. A stochastic representation of the uncertainties could be sufficient. But we argue, that a bias in the upper boundary condition could induce a bias in the estimated parameters. Especially if this bias mainly occurs during the rain event, it could cause a parameter shift during the rain event, similar to the one observed during the closed-eye period. To prevent this, we do estimate the boundary condition. This requires the decreased temporal resolution, which could lead to short-term parameter variations within the rain event. We think that this downside is subordinate to the removal of the bias, especially since it is strongly alleviated by the disspiative soil water movement.

The information about the boundary condition is in the covariances between measurements and the boundary condition, which is highest for the topmost TDR.

We improved the description of the boundary condition in the manuscript (page 8, line 5-11): "The expansion to an augmented state changes the propagation in time. Each component needs an individual forward propagation. We assume the soil hydraulic parameters and Miller scaling factors to be constant in time. This is not possible for the upper boundary condition where the forward equation is unknown from a soils perspective. However, measurements are available to estimate the evaporation and precipitation. Hence, we assume the forward model constant until a new estimation is available. Then, we switch to the estimated boundary condition.

To base the improvement of the upper boundary condition on several measurements, we reduce the temporal resolution of precipitation measurements to change daily and at transitions between precipitation and evaporation. This means, that the upper boundary condition is treated like the parameters, except that the value can change in the forward propagation. The original temporal resolution of the precipitation data (10 minutes) is not required due to the dissipative nature of the Richards equation, which smooths the infiltration front before it reaches the first TDR sensor in a depth of 8.5 cm. The estimation of the averaged boundary condition will ensure that there is no global bias on the parameter estimation during the rain event, but could lead to small short-time parameter drifts within a rain event." and (page 11, line 34): "As we added the boundary condition into our estimation, we can exclude effects due to a bias in the precipitation to cause this parameter shift. The reduced temporal resolution in the boundary condition could only cause short-term parameter changes within the rain event"

*5) A discussion is missing on how a 'closed-eye' EnKF period can be defined in other practical applications. In the paper, a quite heuristic criterion (maximal parameter change within a defined time period) is used which relies on a prior application of the standard EnKF assimilation scheme. Can such a criterion be generalized? The EnKF is usually applied as a pure sequential method without iterations (often in the context of real-time simulations), so the question is whether the 'closed-eye' period can also be defined properly for such cases, or if the iterative EnKF is a prerequisite for the 'closed-eye' EnKF. An additional question is how such a 'closed-eye' period would be defined for a 3D model where precipitation is spatially distributed and the computational demand is much larger than for a 1D soil column.*
**Reply:** In our case the determination of the closed-eye period requires the previous iterative Standard EnKF, but the close-eye period could be possibly determined directly from the current state or the forcing. Such a direct identification then would also enable the application to computationally more expensive models.

We enhanced the discussion accordingly (page 15, line 14-20):"Due to the iterative approach, we could detect the times when the local-equilibrium assumption is violated: the variations of the parameters are larger than the change from initial to final value during later iteration. The changes in the direction of the parameter update then determine the closed-eye period.

Generally, the closed-eye period can be detected, if the operational limits of the model are known. In our case, we base this on the changing parameters, but e.g. a direct detection in the state or forcing could be possible as well.

The Closed-Eye EnKF omits the incorporation of the model structural errors in the parameters and is a generally applicable concept. In this study, it yields better predictions during periods when the underlying assumptions are fulfilled: the drying period after a rain event when there is local equilibrium, showing the strength of the Richards equation there."

*6)* *To my knowledge, the covariance inflation method proposed in Anderson (2009) has not been applied in hydrologic data assimilation so far. It would be interesting to see, how this inflation scheme affects model results, e.g. by providing some information on the calculated inflation factors or by comparison with a standard EnKF assimilation scheme.*
**Reply:** We show the inflation factor during the last iteration of the Standard EnKF and Closed-Eye EnKF in an additional figure now.

**Specific comments:**
*page 1 line 6:* *I think the definition of the 'closed-eye EnKF' should already been given here or at least in the introduction.*
**Reply:** We incorporated it in the introduction (see answer to general comment 1).

*page 2 line 15-16:* *This not fully representative for the study. You should also add*

*your experiments with the standard/ 'closed-eye' EnKF here.*
**Reply:** We extended the introduction (see answer to general comment 1).

*page 3 line 22-25: Please clarify what you define as 'non-equilibrium conditions' in this context. Do you mean preferential flow?*
**Reply:** Preferential flow is one example for a local non-equilibrium condition. Generally, it means that the state does not follow the retention curve. This could also be the case if there is a rapid ground water lift. We added the preferential flow example in the manuscript.

*page 4 line 20-27: The definition of Miller scaling requires more details. How exactly are equations 5+6 incorporated into your model simulations? I guess you define a scaling factor for each model grid cell and use the parameters of the respective soil layer as the "reference material property". Additionally, why does a scaling factor of 1 introduce large uncertainties? This is not obvious from equations 5+6.*
**Reply:** You are correct. We have improved the explanation in the manuscript to:
"A possibility to describe the texture inside soil layers in a simplified way is Miller scaling (Miller and Miller, 1956). They assume geometrical similarity and scale the material properties at each location (grid cell) according to:

$$K(\theta) = K^*(\theta) \cdot \xi^2, \tag{5}$$

$$h_m(\theta) = h_m^*(\theta) \cdot \frac{1}{\xi}, \tag{6}$$

where $*$ denotes the reference material properties of the corresponding layer and $\xi$ is the Miller scaling parameter at this location. This means, that we have to define a Miller scaling factor for each grid cell. The assumption of geometric similarity does not necessarily hold and leads to some uncertainties. The scaling factor themselves are a priori unknown. We assume no heterogeneity (scaling factors of 1), which is most

likely wrong and introduces large uncertainties."

*page 5 line 5-9:* *The TDR measurements for 111 and 136 cm in Figure 4b suggest that there is some dynamics in the lower part of the soil profile. On the contrary, the simulated soil moisture contents are almost constant. This is an indication that there might be some problems with the assigned lower boundary condition. The constant simulated soil moisture values are probably an effect of the sand layer that was introduced as layer 5 which acts as a hydraulic barrier to the upper soil profile. You could try to use the soil hydraulic parameters from layer 4 also for layer 5 to see, if the simulated soil moisture dynamics can be improved in the lower part of the soil profile. Additionally, it is not clear why you have chosen a water table depth of 4m. Is this value based on water level data from surrounding wells?*

**Reply:** Considering the noise in the sensor, we do not see dynamics in the TDR measurements at 111 and 136 cm depth. Additionally, the measurements are in an hourly interval, but we interpolated linearly when there was missing data leading to the linear stretches. We have added this information to the Figure.

We agree, that in case of dynamics, we should use the soil parameters of layer 4 for layer 5 as well. Without dynamics, we chose the sand instead to decouple the from the ground water table. Because of this decoupling, the position of the groundwater table doesn't matter. We do not have measurements of the ground water table and chose the depth of 4m to be consistent with Wollschläger et al. (2009).

*page 7 line 25:* *If the soil moisture measurements exceed the calibrated saturated water, why don't you use the data from the volumetric soil samples (page 5 line 24) as an initial guess?*

**Reply:** The data from volumetric samples is recorded for each TDR location individually, but we still assume a homogeneous reference material for the complete layer with only one saturated water content value. Furthermore, changing the saturated

water content affects the material properties directly. We did not want to alter the material properties estimated by Wollschläger et al. (2009) strongly and thus, chose to estimate the saturated water content instead, to keep the change as small as possible. As long as we do not actually reach saturation, the value is not important.

*page 8 line 9-11: Does this mean that you only use daily averages of precipitation? In the data assimilation experiments rather short time periods are investigated and the simulations are compared to hourly soil moisture measurements. So it would make much more sense to also use atmospheric forcing data with a similar time resolution which should be available from the weather station (page 3 line 3-4).*
**Reply:** In this case the precipitation is averaged over roughly 5 hours during day 17 and 8 hours during day 18. The reduced temporal resolution is necessary to estimate the boundary condition (see answer to general comment 4). Additionally, for the evaporation the estimation of daily values has a smaller uncertainty than the estimation of hourly values (Foken, 2006).

*page 8 line 30-31: Which value for the cut-off radius did you use in the 5th order polynomial for soil water content?*
**Reply:** We empirically chose a very large value of 153 cm for the cut-off radius (corresponds to $2c$ in the 5th order piecewise function). We added this information in the manuscript.

*page 9 line 2-4: Why is the localization different for hydraulic conductivity and Miller scaling factors? Miller scaling factors are strongly related to hydraulic conductivities and should therefore also have a similar localization.*
**Reply:** The Miller scaling factors have a local influence at the corresponding position of the TDRs in each layer, while $K_0$ is defined for the complete layer. Because of that we applied a stricter localization to the scaling factors.

*page 9 line 9-11:* *Did you also use localization in the derivation of the inflation factors? This is necessary because otherwise uncertainties are increased in model areas that are not updated by the EnKF.*
**Reply:** Yes.

*page 10 line 20-23:* *How are initial Miller scaling factors perturbed?*
**Reply:** The uncertainty of the natural logarithm was set to 0.4. We added this information in the manuscript.

*page 11 line 7-11:* *It is unclear how the initial conditions for time period C were exactly created. How were the final (updated) states from time period B re-sampled from 40 to 100 ensemble members? Did you perturb the ensemble mean of Miller scaling factors from time period A or each of the 40 ensemble members individually? Was this perturbation constant in space or was the perturbation on the grid cell/ soil horizon level? Please also take into account that this perturbation creates inconsistencies between states and parameters which could be alleviated by a spin-up period.*
**Reply:** We used 100 ensemble members for time period B (we added this in the manuscript). Thus, we could use the exact states as initial conditions. For the Miller scaling factors we only used the mean value. The block diagram now shows these details. As we added the Miller scaling factors at measurement locations to the augmented state, we perturbed the values there. We are aware that the perturbations destroy previously acquired correlations, but decided not to use a spin-up phase (see answer to general comment 3).

*page 11 line 32 - page 12 line 4:* *It is not obvious from Figure 5 whether hydraulic conductivity reaches its initial value by the end of time period C. Please be more*

*quantitative here: How large is the change within the 'closed-eye' period and how large is the difference between initial and final values? It would also be good to show how the update of hydraulic conductivity evolves over the different iterations of the EnKF. Is there a continuous increase of this parameter during the iterations? Additionally, the increase of hydraulic conductivity during the closed-eye period could also be related to the adaptive covariance inflation. The calculation of the inflation factors is influenced by the mismatch between simulations and measurements. As this mismatch increases during the infiltration event, this could also lead to higher inflation factors which increase the Kalman gain and thus the parameter update for this period. Did you check how the inflation factors evolve over this time period? Maybe it's also worthwhile to repeat these simulations without covariance inflation in order to exclude possible artefacts from the inflation scheme.*

**Reply:** We added the temporal evolution during the 10 iterations to the Figure. The observed changes in the parameters cannot be caused by the covariance inflation. We agree, it is possible that the inflation could cause parameters to change stronger during the infiltration, but this does not explain the changing directions, on which we base the closed-eye period. The maximal inflation factor $\lambda$ during the last iteration of the Standard EnKF was 1.013. The values are larger during the Closed-Eye EnKF when the mismatch is larger during the infiltration. We added a new Figure with the temporal evolution during the last iterations in the manuscript.

*page 12 line 14-17: The 'closed-eye' EnKF experiments are added on top of the standard EnKF experiments with an additional parameter perturbation. This makes it quite difficult to compare these two experiments. I suggest to use exactly the same model set-up for both experiments, i.e. to repeat the 'closed-eye' EnKF experiment with the same initial conditions as the experiment with standard EnKF. Otherwise, a direct comparison of both methods is not possible. An additional question is why the 'closed-eye' EnKF and the parameter perturbation is only performed for the first soil layer.*
**Reply:** We did not see changing parameters in the second layer. Thus, we only applied the closed-eye period on the first layer. We do not want to change the work flow in the three stages. Instead, we now compare the results from the Closed-Eye EnKF to a Standard EnKF with 20 iterations to achieve a fair comparison.

*page 12 line 23-24: What are the 'believed true material properties'? Given the fact that this is a real-world experiments, where the true material properties are unknown, this is quite speculative.*
**Reply:** We agree, there are no true material properties, but there are, what we call 'believed true material properties', which are only valid in the observed range. We have added our definition (page 12, line 24): "This means, if the parameters are constant in time, we believe, that the parameters can represent reality in the observed water content range during times, when the underlying assumptions hold."

*page 13 line 13-22: Please be more quantitative here by providing performance measures such as root mean square error or Nash-Sutcliffe efficiency.*
**Reply:** We deliberately did not include a quantitative measure here. We wanted to focus on the qualitative result, whether the model can describe the data or not. We think this result can be seen directly. Especially, since the quantitative result of these measures can be easily misinterpreted. The root mean square error is only meaningful, if the model can represent the data with statistically independent errors, which is not the case here.

*page 13 line 26-30: In Figure 10b the simulated ensemble mean for each layer is always above the measured soil moisture values at the end of time period C. Why does this offset not appear in the beginning of time period D in Figure 10d? Shouldn't period D start with the simulation results from period C?*
**Reply:** We start with the last states from the last iteration from the Standard EnKF or

Closed-Eye EnKF respectively, as they represent the best possible estimation for the initial condition for time period D. The results from the free run in the time period B and C already have a bias and consequently should not be used. For time period B and C we performed an EnKF with state estimation to achieve a good initial condition. We added the information in the manuscript.

*page 14 line 28-31:* *You should also discuss how such a 'closed-eye' period can be defined in practical applications.*
**Reply:** (See answer to general comment 5.)

*Figure 2:* *Why is there no evaporation flux in time period A and at the beginning of time period B? If is is a dry period, there should also be evaporation.*
**Reply:** Due to the previous dry period of over one month, the evaporation will be far below the evaporation during wet conditions and almost 0.

**Technical corrections:**
**Reply:** We made the corrections.

Please also note the supplement to this comment:
http://www.hydrol-earth-syst-sci-discuss.net/hess-2016-296/hess-2016-296-AC2-supplement.pdf